


# HMRFS-TP: long-term daily gap-free snow cover products over the Tibetan Plateau from 2002 to 2021 based on Hidden Markov Random Field model

Yan Huang[1,2+*], Jiahui Xu[1,2+], Jingyi Xu[1,2], Yelei Zhao[1,2], Bailang Yu[1,2], Hongxing Liu[3], Shujie Wang[4], Wanjia Xu[1,2], Jianping Wu[1,2], Zhaojun Zheng[5]

[1]Key Laboratory of Geographic Information Science, Ministry of Education, East China Normal University, Shanghai 200241, China
[2]School of Geographic Sciences, East China Normal University, Shanghai 200241, China
[3]Department of Geography, the University of Alabama, Tuscaloosa, AL 35487, USA
[4]Department of Geography, Earth and Environmental Systems Institute, Pennsylvania State University, University Park, PA 16802, USA
[5]National Satellite Meteorological Center, Beijing 100081, China

[+]These authors contributed equally to this work.
*Correspondence to*: Yan Huang (yhuang@geo.ecnu.edu.cn)

**Abstract.** Snow cover plays an essential role in climate change and the hydrological cycle of the Tibetan Plateau. The widely used Moderate Resolution Imaging Spectroradiometer (MODIS) snow products have two major issues: massive data gaps due to frequent clouds and relatively low estimate accuracy of snow cover due to complex terrain in this region. Here we generate long-term daily gap-free snow cover products over the Tibetan Plateau at 500 m resolution by applying a Hidden Markov Random Field (HMRF) technique to the original MODIS snow products over the past two decades. The data gaps of the original MODIS snow products were fully filled by optimally integrating spectral, spatiotemporal, and environmental information within HMRF framework. The snow cover estimate accuracy was greatly increased by incorporating the spatiotemporal variations of solar radiation due to surface topography and sun elevation angle as the environmental contextual information in HMRF-based snow cover estimation. We evaluated our snow products, and the accuracy is 98.31% in comparison with *in situ* observations and 92.44% in comparison with high-resolution snow maps derived from Landsat-8 imagery. Our evaluation also suggests that the incorporation of spatiotemporal solar radiation as the environmental contextual information in HMRF modelling, instead of the simple use of surface elevation as the environmental contextual information, results in the accuracy of the snow products increase by 2.72% and the omission error decrease by 4.59%. The accuracy of our snow products is especially improved during snow transitional period and over complex terrains with high elevation as well as sunny slopes. The new products can provide long-term and spatiotemporally continuous information of snow cover distribution, which is critical for understanding the processes of snow accumulation and melting, analysing its impact on climate change, and facilitating water resource management in Tibetan Plateau. This dataset can be freely accessed from the National Tibetan Plateau Data Center at https://doi.org/10.11888/Cryos.tpdc.272204 (Huang and Xu, 2022).





## 1 Introduction

Snow cover has the characteristics of low thermal diffusivity, high reflectivity, and strong water storage capacity, which has a profound effect on climate change (Gao et al., 2012), radiation budget (Yang et al., 2014; Huang et al., 2019), hydrological cycle (Dong, 2018), and human activities (Cereceda-Balic et al., 2020). The Tibetan Plateau (TP) has abundant snow cover, with highest elevation and largest snow cover area in the middle latitudes of Northern Hemisphere (Qiu, 2008; Yao et al., 2019). Snow is highly sensitive to climate change (Chen et al., 2018), and snowmelt water quantity is closely connected to

the supply of soil moisture on the plateau (Wang et al., 2018) and the runoff of numerous rivers (Immerzeel et al., 2010). Long-term and detailed snow cover information is fundamental to investigating climate change and hydrological cycle of TP. Remote sensing has allowed extraction of historical and near-real-time snow cover extent over inaccessible areas (Yang et al., 2015; Li et al., 2018). Snow products could be acquired via remote sensing satellites, such as the Landsat and Sentinel-2 series. Although these satellites have a high spatial resolution, they have a relatively coarse (10 or 16 days) temporal

resolution, thereby rendering them insufficient to monitor the temporal variations in snow cover (Huang et al., 2022). Moderate Resolution Imaging Spectroradiometer (MODIS) has commenced producing snow products at 500 m resolution from 2000, which have been widely utilized as the primary datasets for monitoring snow cover (Muhammad and Thapa, 2020; Kilpys et al., 2020). The accuracy of MODIS snow products is greater than 85% at the global scale under clear sky (e.g., Parajka et al., 2012; Yang et al., 2015). However, these products have many data gaps due to frequent clouds, causing

discontinuity in the time and space of the products (Liu et al., 2020b). In addition, the complex terrain of TP makes it more challenging for accurate snow cover detection. Previous studies (Dong and Menzel, 2016; Dariane et al., 2017) have suggested the low accuracy of MODIS snow products for mountainous areas.

Various data-gap-filling techniques have been proposed to produce seamless MODIS snow cover products including multi-source combination, temporal, spatial, and spatiotemporal filters (Xiao et al., 2021; Hussainzada et al., 2021; Richiardi et al.,

2021). The multi-source combination method combines MODIS with passive microwave sensor data (Li et al., 2019b; Li et al., 2020). Although this method can be used to fill most data gaps, the accuracy of the combined product depends more on the accuracy of the microwave sensor data, and the spatial resolution (>1 km) is not sufficient to meet the demands for accurately assessing the snow cover variability (Wang et al., 2009). Many attempts have been directed toward filling the gaps in MODIS optical remote sensing data based on spatial and temporal information. Temporal methods are used to

reclassify the data-gap pixels by inferring the land cover types of the current pixels under clear sky within a few days before or after (e.g., the previous or next day) (Li et al., 2019a; Tran et al., 2019). Spatial methods estimate data-gap pixels based on gap-free pixels in the spatial neighborhood (Hou et al., 2019). Relevant environmental information (e.g., snow lines and topography) has been introduced into spatial methods (Wang et al., 2019; Kilpys et al., 2020). Spatiotemporal methods have been integrated to fill as many data gaps as possible (Parajka and Blöschl, 2008; Kilpys et al., 2020). These studies

integrated all information (i.e. spatiotemporal, and environmental information) according to heuristic rules instead of rigorous quantitative model. Huang et al. (2018) developed a Hidden Markov Random Field (HMRF) framework to





optimally combine all information. This technique not only fills data gaps but also provides fine improvement in accuracy compared to the original MODIS snow products.

The long-term series and high-precision products are the basis for snow cover research on the TP, which enable the
monitoring and analysis of snow cover phenology and more comprehensive understanding of the snow cover trend. However, the existing daily products for the TP have an earlier end date (the latest version at 500 m resolution product ended in 2015), and some products still have a small number of data gaps (Tang et al., 2013; Yu et al., 2016; Qiu et al., 2017; Zheng and Cao, 2019). Thus, long-term and high-precision daily snow cover products for the TP should be generated using reliable data-gap-filling methods.

Here we generate long-term daily snow cover products over the TP by applying a Hidden Markov Random Field (HMRF) technique (Huang et al., 2018) to the original MODIS snow products from 2002 to 2021. In the previous HMRF modelling technique (Huang et al., 2018), surface elevation was considered as a surrogate for environmental information in mountainous regions. However, according to *in situ* photos from field survey in the TP, we found that the distribution of snow cover differed from the model results, even at the same elevation level (Figs. 1b and 1c). The snow on the TP is
strongly infected by complex topography (e.g., slope, aspect, sunlight duration, and solar incidence angle). Thus, while generating our new snow cover product, we incorporated solar radiation as a comprehensive indicator of topographical effects to correct the snow identification errors caused by the complex terrain of the TP.

Our study is outlined as follows: first, we present our study area and datasets. Then, we present the HMRF framework and data-processing flowchart. By employing this modelling technique to the TP, we produced daily gap-free snow cover
products for the TP from 2002 to 2021. We also validated our new snow products against *in situ* observations, snow cover mapped from Landsat-8 imagery, and snow cover data estimated from the initial MODIS and original HMRF modelling technique.

## 2 Study area and data

### 2.1 Study area

The TP is situated in western China, spanning 26°00′-39°46′N and 73°18′-104°46′E (Fig. 1). It encompasses approximately $2.56 \times 10^6$ km², with an average altitude exceeding 4000 m, and is one of the most susceptible regions to climate change (Jing et al., 2019). The air temperature in the TP increased from the northwest to southeast, with more precipitation in the southeast and less in the northwest (Wang et al., 2018). Generally, the TP has comparatively low temperatures and has largest distribution of glaciers and snow in China (Liang et al., 2017).

The status of the snow in the TP changes rapidly as a result of multiple factors, such as temperature, precipitation, synoptic forcing, and large-scale ocean-atmosphere oscillations (You et al., 2020), which may lead to sublimation and snowfall (Li et al., 2020). Affected by the Indian Ocean monsoon and East Asian summer monsoon, the southeast TP has plenty water vapor, which results in a large amount of cloud, particularly in spring and summer (Yang et al., 2015).



**Figure 1: Topography, meteorological stations, and survey photos of the TP. (a) Surface elevation and distribution of meteorological stations in the TP. Landsat-8 imagery utilized for validation and sample area are also shown. (b) and (c) *in situ* photos from field survey in the TP.**

## 2.2 Datasets

### 2.2.1 Daily MODIS snow cover products

We used version 6 MODIS daily snow products on Terra (MOD10A1 v6) and Aqua (MYD10A1 v6) from May 15, 2002, to December 31, 2021 (Hall and Riggs, 2016a; Hall and Riggs, 2016b). Google Earth Engine (GEE) (https://earthengine.google.com) platform was used for pre-processing the MODIS snow product. We used MODIS/006/MOD10A1 and MODIS/006/MYD10A1 datasets and NDSI_Snow_Cover and NDSI_Snow_Cover_Class bands. NDSI_Snow_Cover represents the value of the Normalized Difference Snow Index (NDSI), ranging 0–100. The



values in the NDSI_Snow_Cover_Class band include 200, 201, 211, 237, 239, 250, 254, and 255, which represent "missing data", "no decision", "night time", "inland water", "ocean", "clouds", "detector saturated", and "filled data", respectively (Riggs et al., 2019).

The original NDSI data were reclassified as snow, non-snow, and data-gaps classes (Huang et al., 2018). The pixels with an NDSI value of 40–100 in the NDSI_Snow_Cover band were reclassified as snow (Riggs et al., 2017). The pixels with an

NDSI value of 0–40 in the NDSI_Snow_Cover band and with corresponding values of 0, 211, 237, and 239 in the NDSI_Snow_Cover_Class band were reclassified as non-snow. The remaining pixels were reclassified as data-gaps. Then, we combined the MOD10A1 and MYD10A1 reclassification results of the same day according to the following rules: when a pixel included gap-free data in both MOD10A1 and MYD10A1 products, value in the MOD10A1 product was used; when a pixel included gap-free only in the one product, the gap-free data value was used. Finally, the combined Terra and Aqua

results were re-projected onto Universal Transverse Mercator (UTM zone 45) at 500 m resolution, which was used as the initial snow cover products for filling data gaps in this research.

We identified the data gaps in original composite MODIS products, and found the annual average data gap proportion of the TP was 33.63%–38.75%, with an average of 36.12% (Fig. 2). The monthly average data gaps are also shown in Fig. 2, and found that the data gaps of the TP were the largest in summer (June to August), with an average of 45.20%. The data gaps

declined rapidly starting from September and were the smallest in November (with an average of 28.00%), and gradually increased in winter and spring.

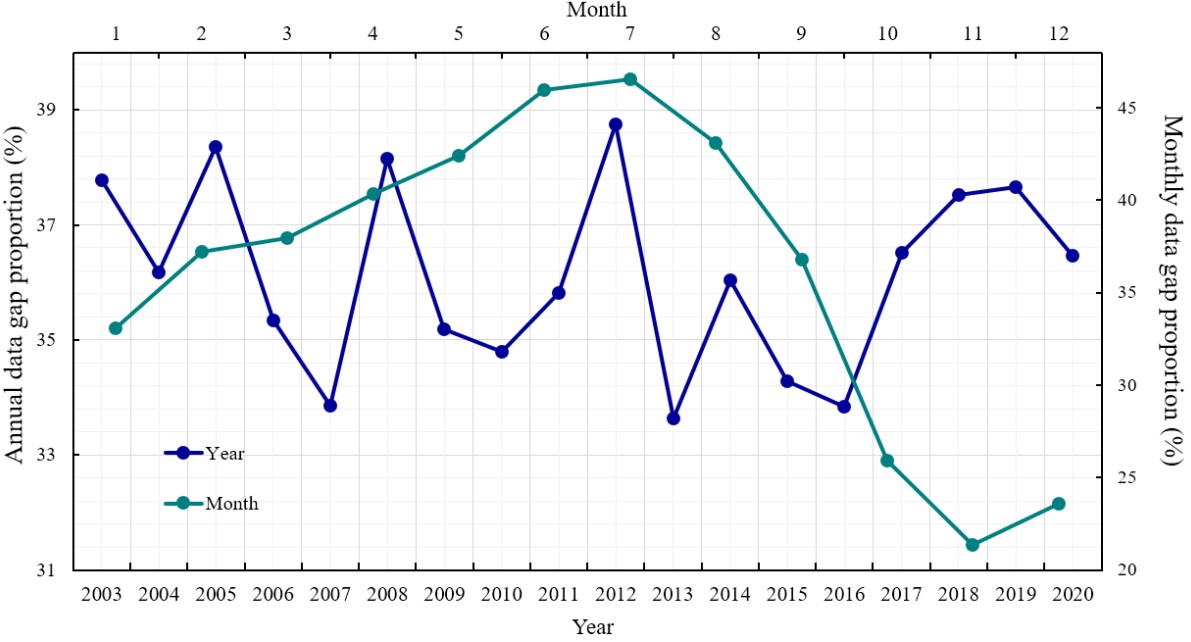

**Figure 2: Annual and monthly average proportion of data gaps of original composite MODIS snow products.**





### 2.2.2 Digital Elevation Models (DEMs)

We used the Shuttle Radar Topography Mission (SRTM) DEMs to calculate the topographical parameters (e.g., elevation, slope, and aspect). The SRTM DEM is at 90 m resolution in GeoTIFF format was from the United States Geological Survey (USGS). We then pre-processed the DEM data, including mosaicking, reprojection, resampling, and clipping.

### 2.2.3 *In situ* observations

We used snow depths from 137 *in situ* stations in the TP (Fig. 1a) obtained from October 1, 2002, to March 31, 2021, which
were provided by the National Meteorological Centre of China. Fig. 1a shows all stations over the TP, which record coordinates, observation time, snow depth, and snow pressure. The surface elevation of these stations ranges from 1187 m to 5310 m, with an average of 3240 m. Overall, 72% stations are situated in the east (>95°E), and only 22% are situated in high-altitude areas (>4000 m). A 3 cm threshold was utilized to evaluate the snow classes using *in situ* snow depth (Huang et al., 2022). The snow depth was reclassified as no-snow if it was less than 3 cm; if not, it was considered as snow.

### 2.2.4 Landsat-8 imagery

The snow cover mapped from Landsat-8 imagery were used to validate and evaluate our snow cover product. Using the GEE cloud platform, we selected 62 Landsat-8 images with less than 10% cloud coverage under clear sky for verification. The detailed Landsat-8 imagery information is shown in the supplementary material (Table A1). The selected images were distributed throughout the study area from January 1, 2016, to December 31, 2018. Using the SNOWMAP method proposed
by Hall et al. (1995), we generated a snow cover map using Landsat-8 imagery for the TP. The SNOWMAP algorithm classified the pixels with NDSI values > 0.4, green band > 0.10, and SWIR1 band > 0.11 as snow (Huang et al., 2011). To improve the accuracy of resampling 30 to 500 m resolution, we adopted the following steps for resampling: first, for each MODIS pixel, we calculated the amount of Landsat snow-covered pixels in the current MODIS pixel. Second, we divided the snow-covered Landsat pixels by sum of the Landsat pixels contained in each MODIS pixel (a 500 m MODIS pixel is
close to 277 Landsat pixels) (Crawford, 2015; Liu et al., 2020a). Finally, we reclassified pixels whose results in the previous step were less than 0.5 as no-snow; otherwise, they were reclassified as snow. Hence, we obtained resampled 500 m resolution Landsat images for further verification.

### 3 HMRF modelling technique

The HMRF framework optimally combines MODIS spectral, spatiotemporal, and environmental information to fill data gaps,
thereby increasing snow estimate accuracy (Huang et al., 2018). This framework is expressed as a linear energy function in which the total energy is modelled as the combination of all information (Eq. (1)). Thus, the HMRF framework requires to specify energy function for each information and to determine the optimal parameters that minimize the total energy.



$$U_T(\beta_n, x_i, N_{sp}, N_{tp}, I_{ev}) = \lambda_{xi} U_{xi}(\beta_n, x_i) + \lambda_{st} U_{st}(\beta_n, N_{sp}, N_{tp}) + \lambda_{ev} U_{ev}(\beta_n, I_{ev}) \tag{1}$$

where $\lambda_{xi}, \lambda_{st}$, and $\lambda_{ev}$ are the spectral, spatiotemporal, and environmental parameters, respectively; and $U_{xi}, U_{st}$, and $U_{ev}$ are

the spectral, spatiotemporal, and environmental energy functions, respectively.

In the original HMRF modelling technique, Huang et al. (2018) used surface elevation to represent the environmental association in mountainous areas (hereafter referred to as $HMRF_{ele}$). However, because of the spatial heterogeneity of the TP, using only relative elevation cannot reflect the influence of complex terrain on snow cover (Figs. 1b and 1c). Therefore, the influence of additional topographic factors (e.g., slope, aspect, sunlight duration, and solar incidence angle) on snow cover

must be considered. Based on the above reasons, we incorporated solar radiation as a surrogate for environmental information in HMRF framework (hereafter referred to as $HMRF_{solar}$). Further details are provided in Section 3.3.

The overall flowchart of the $HMRF_{solar}$ modeling technique is shown in Fig. 3. We used daily composite MODIS snow cover products as initial dataset. First, we calculated the optimal parameters of each information using the Ho-Kashyap and DFBETAS algorithms (Bormann et al., 2012), based on randomly divided training samples on the initial snow cover product.

Second, we determined the snow and non-snow classes of each pixel in the initial snow cover products using the optimal parameters and $HMRF_{solar}$ algorithm. In the first round of the $HMRF_{solar}$ algorithm, we employed a $3 \times 3 \times 3$ spatiotemporal cubic neighborhood to model spatiotemporal energy. If data gaps remained, we further expanded the cubic neighborhood of these pixels. Finally, we validated and evaluated our $HMRF_{solar}$-based products against the *in situ* observations, snow mapped from Landsat-8 imagery, and snow cover data from the initial MODIS and original $HMRF_{ele}$

modelling technique.

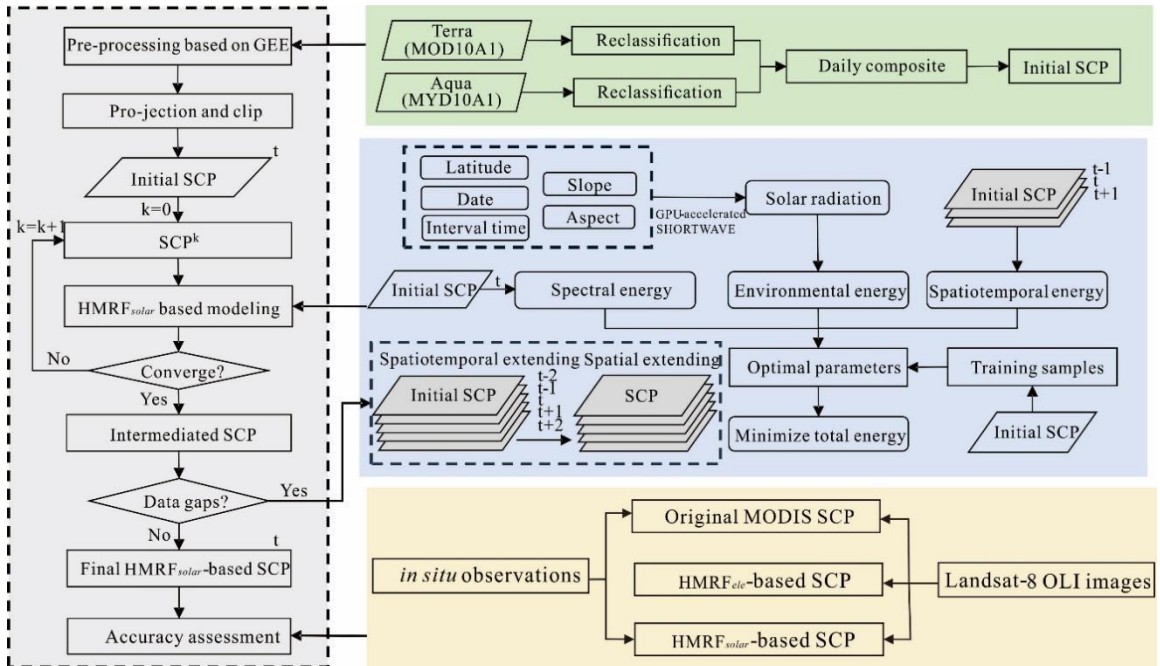

**Figure 3: Overall flowchart of the $HMRF_{solar}$-based framework. (SCP stands for snow cover products.)**



### 3.1 Spectral information

The spectral energy is considered as the possibility of the pixel pertaining to non-snow or snow according to its spectral
information. Fractional Snow Cover (FSC) represents the proportion of snow in a pixel, which can be calculated using the
NDSI value provided by the MODIS snow product (Salomonson and Appel, 2004). Aqua Band 6 was reinstated using an
image restoration algorithm in MODIS version 6 (Gladkova et al., 2012); thus, a general equation can be used to calculate
the FSC. The spectral probability $P(x_i|\beta_1)$ of snow class can be defined as follows:

$$P(x_i|\beta_1) = (-0.01 + 1.45 \times NDSI)/100 \tag{2}$$

The spectral energy $U_{xi}(\beta_n, x_i)$ can be calculated as follows:

$$U_{xi}(\beta_n, x_i) \begin{cases} -[P(x_i|\beta_1)], \text{if } n = 1 \\ -[1 - [P(x_i|\beta_1)], \text{if n} = 2 \end{cases} \tag{3}$$

where if pixel $i$ is classified as no-snow class, then $n = 2$; if pixel $i$ is the snow class, $n = 1$.

### 3.2 Spatiotemporal information

To fully take advantage of spatiotemporal information, we combined the temporal and spatial information to form a
spatiotemporal cubic neighborhood $N_{st}$ (Huang et al., 2018). In modeling, initial snow product is updated iteratively using
classification results from the previous iteration. Convergence ends when the proportion of pixels whose type changes
between two subsequent iterations is less than 0.1%, in which case the data gaps need to be calculate. If the data gaps still
exist, the cubic neighborhood is temporally and spatially expanded next round.

We primarily set a $3 \times 3 \times 3$ cubic neighborhood; that is, the current pixel and its neighboring pixels formed a cube on day $t$,
the day before $(t - 1)$, and the day after $(t + 1)$. In this condition, the energy equation of the current pixel is expressed as the
weight of the proportion of snow and non-snow in the spatiotemporal neighborhood of the current pixel. The weight is
reversely proportional to the distance from the neighboring pixel to the center pixel:

$$U_{st}(\beta_n, N_{sp}, N_{tp}) = \begin{cases} -\sum_{x=-1}^{1}\sum_{y=-1}^{1}\sum_{t=-1}^{1} \frac{\frac{S(x,y,t)}{Dist(x,y,t)}}{\frac{V(x,y,t)}{Dist(x,y,t)}}, \text{if n} = 1 \\ -\sum_{x=-1}^{1}\sum_{y=-1}^{1}\sum_{t=-1}^{1} \frac{\frac{NS(x,y,t)}{Dist(x,y,t)}}{\frac{V(x,y,t)}{Dist(x,y,t)}}, \text{ if n} = 2 \end{cases} \tag{4}$$

where $(x, y, t)$ is the relative coordinates of the pixel in the spatiotemporal cube; $S(x,y,t)$, $NS(x,y,t)$, $V(x,y,t)$,
and $Dist(x,y,t)$ are defined respectively as follows:

$$S(x,y,t) = \begin{cases} 1, \text{if } (x,y,t) \text{ pertains to snow} \\ 0, \text{ if } (x,y,t) \text{ pertains to non} - \text{snow} \end{cases} \tag{5}$$

$$NS(x,y,t) = \begin{cases} 1, \text{ if } (x,y,t) \text{pertains to non} - \text{snow} \\ 0, \text{ if } (x,y,t) \text{pertains to snow} \end{cases} \tag{6}$$

$$V(x,y,t) = \begin{cases} 1, \text{ if}(x,y,t) \text{ pertains to valid class} \\ 0, \text{ if } (x,y,t) \text{ pertains to data gaps} \end{cases} \tag{7}$$

$$Dist(x,y,t) = \sqrt{x^2 + y^2 + wt^2} \tag{8}$$





where $Dist(x,y,t)$ is the distance from each neighborhood pixel to the center pixel, which determines the weight of each area pixel; $w$ denotes the weight of the temporal distance related to the spatial distance. This value was calculated via semi-variogram analysis and was determined as 3 in this study.

If data gaps still remain in $3 \times 3 \times 3$ spatiotemporal cubic neighborhood, we further expanded spatiotemporal neighborhood to $5 \times 5 \times 5$. In this study, when we used a $5 \times 5 \times 5$ cubic neighborhood, the data gaps were reduced to 0.0142%, which

had little effect on snow cover estimate accuracy. Hence, the temporal neighborhood remained at 5, whereas the spatial neighborhood continued to expand. Finally, all data gaps could all be filled by continuously expanding the spatial neighborhood, which remarkably improved the MODIS estimate accuracy.

### 3.3 Environmental contextual information

The geographic location and seasons can determine the total solar radiation received on the TP, and the complex topography

of the TP determines the availability of solar radiation at specific locations in the region, which further affects accumulation and melting of snow cover and determines its distribution (You et al., 2020). Here we use solar radiation to include the influence of environmental factors on snow cover, such as slope, aspect, sunlight duration, and solar incidence angle.

Kumar et al. (1997) originally proposed the solar radiation model, SHORTWAVE, to calculate the direct, reflected, and diffuse solar energy received by the ground. Huang et al. (2015) used a Graphics Processing Unit (GPU) methodology to

improve the speed and effectiveness for modelling. In our study, we used the GPU-accelerated SHORTWAVE model as the environmental parameter in the $\text{HMRF}_{solar}$ model to calculate the daily solar radiation of the TP.

The GPU-accelerated SHORTWAVE model uses latitude, slope, aspect, date, and interval time as inputs. To calculate the overall solar radiation in a day, we subdivided the day into subsequent time intervals, and set 15 min as the suitable interval time in this paper (Kumar et al., 1997; Antonic, 1998). The overall solar radiation $I_a$ at time $t$ can be calculated using Eq. (9):

$I_a = I_p + I_d + I_r$ (9)

where $I_p$, $I_d$, and $I_r$ are the direct, diffuse, and reflected solar radiation, respectively, and can be estimated using the following equations:

$I_p = I_0 \cdot \tau_b \cdot cos\theta$ (10)

$I_d = \frac{I_0 \cdot \tau_d \cdot cos^2 \beta}{2 sin\alpha}$ (11)

$I_r = \frac{r \cdot I_0 \cdot \tau_r \cdot sin^2 \beta}{2 sin\alpha}$ (12)

where $I_0$ is the extra-atmospheric solar flux; $\tau_b$ denotes the atmospheric transmittance; $\tau_d$ denotes the diffuse skylight transmittance; $\tau_r$ is the reflectance transmissivity (Gul et al., 1998); $r$ is set to 0.2 in the current paper(Kumar et al., 1997); $\alpha$ is the solar elevation angle, which varies in terms of the latitude, season and time; $\theta$ is the solar incidence angle, which is determined by solar elevation angle $\alpha$, the solar azimuth angle, surface slope ($\beta$) and aspect (Huang et al., 2015). The effect





of the geographic location and seasons on the total solar radiation are represented by solar elevation angle variable $\alpha$, and the

effect of the complex topography on the total solar radiation is represented by surface slope ($\beta$) and aspect variables.

We repeated the calculation for each time interval. Last, we accumulated the solar radiation from sunrise to sunset to acquire

the daily values.

Areas that receive more solar radiation generally experience later snow accumulation and earlier snowmelt than areas that

receive less solar radiation. Therefore, if pixels in a spatial neighborhood have higher solar radiation and the classification

result is snow, the probability that the center pixel is also snow increases. The environmental energy equation is defined as

follows:

$$U_{ev}(\beta_n, N_{tp}, N_{sp}) = \begin{cases} -\sum_{x=-1}^{1}\sum_{y=-1}^{1}\frac{SE(x,y)}{SE(x,y)+NSE(x,y)}, & \text{if } n = 1 \\ -\sum_{x=-1}^{1}\sum_{y=-1}^{1}\frac{NSE(x,y)}{SE(x,y)+NSE(x,y)}, & \text{if } n = 2 \end{cases} \tag{13}$$

where $(x,y)$ are coordinates of the pixel in the spatial neighborhood $N_{sp}$; $SE(x,y)$ and $NSE(x,y)$ are defined as follows:

$$SE(x,y) = \begin{cases} 1, \text{if } (x,y) \text{ belongs to snow and } solar(x,y) \geq solar(i) \\ 0, \text{others} \end{cases} \tag{14}$$

$$NSE(x,y) = \begin{cases} 1, \text{if } (x,y) \text{ belongs to non}-\text{snow and } solar(x,y) \leq solar(i) \\ 0, \text{ others} \end{cases} \tag{15}$$

where $solar(x,y)$ and $solar(i)$ represent the solar radiation received by the neighboring pixels and central pixel,

respectively.

## 4 Results

### 4.1 Accuracy assessment based on *in situ* observations

Using $HMRF_{solar}$-based method, we produced gap-free snow cover products over the TP with a daily 500 m resolution from

May 15, 2002, to December 31, 2021. We first compared our daily $HMRF_{solar}$-based and original MODIS snow products

with snow depth observed from 137 *in situ* stations from 2002–2021 (Table 1). The overall accuracy (OA) of all gap-free

pixels in original MODIS products was 97.96%. After employing the $HMRF_{solar}$-based method, the OA of these gap-free

pixels was increased to 98.31%. The omission error was reduced by 0.57%. Since the *in situ* stations in the TP are primarily

distributed in low-altitude areas (Fig. 1a), only using *in situ* observations may not be representative and lead to biases in the

accuracy assessment. Therefore, we also conducted evaluation in comparison with snow cover mapped from Landsat-8

imagery obtained from 2016–2018. We selected Landsat-8 imagery acquired during 2016–2018 for two reasons. First, Fig. 2

shows the proportion of data gaps in the original MODIS products in these three years covered low, medium, and high

values, which could represent the range of data gaps in original products covering 2002–2021. Second, according to our

accuracy evaluation of $HMRF_{solar}$-based snow products in comparison with *in situ* observations from 2016–2018 (Table 2),

we found that the OA of these three years was 97.76%, which is nearly the same as the OA from 2002–2021 (Table 1). This



suggests that the accuracy from 2016–2018 of the products can represent that of the long-term series products. Therefore, in the follow-up accuracy evaluation, we used the snow cover mapped from the Landsat-8 imagery from 2016–2018 as the true verification value.

**Table 1.** Confusion matrices between $HMRF_{solar}$-based, original MODIS snow products, and *in situ* observation for gap-free pixels in original MODIS snow products during 2002–2021.

| *In situ* observation | $HMRF_{solar}$-based snow products | | | Original MODIS snow products | | |
|---|---|---|---|---|---|---|
| | Snow | Non-snow | Total | Snow | Non-snow | Total |
| Snow | 4766 (66.61%) | 2389 (33.39%) | 7155 | 4725 (66.04%) | 2430 (33.96%) | 7155 |
| Non-snow | 3350 (1.01%) | 328110 (98.99%) | 331460 | 4493 (1.36%) | 326927 (98.64%) | 331460 |
| Total | 8116 | 330499 | 338615 | 9218 | 329397 | 338615 |
| Overall accuracy | 98.31% | | | 97.96% | | |

**Table 2.** Confusion matrices between $HMRF_{solar}$-based snow products and *in situ* observation during 2016–2018.

| *In situ* observation | $HMRF_{solar}$-based snow products | | |
|---|---|---|---|
| | Snow | Non-snow | Total |
| Snow | 1325 (43.15%) | 1746 (56.85%) | 3071 |
| Non-snow | 1488 (1.06%) | 139501 (98.94%) | 140989 |
| Total | 2813 | 141247 | 144060 |
| Overall accuracy | 97.76% | | |

## 4.2 Accuracy assessment based on Landsat-8 imagery

A total of 62 Landsat-8 images (Fig. 1a, Table A1) with less than 10% cloud coverage were selected for this accuracy assessment. We constructed the confusion matrices for the $HMRF_{solar}$-based snow cover products, $HMRF_{ele}$-based snow products, and original MODIS snow cover products against snow cover mapped from Landsat-8 imagery products during 2016–2018 (Table 3). The OA for gap-free pixels in original MODIS products was 89.45%. When we used surface elevation to represent the environmental association in HMRF modelling technique, the OA of these gap-free pixels was increased to 89.72%. Since using surface elevation only cannot well represent the influence of complex topography on snow cover distribution over the TP, this improvement is still limited. After incorporating the solar radiation to model the comprehensive influence of multiple topographic factors on snow cover, the OA of our $HMRF_{solar}$-based snow products was increased by 2.99% compared with the OA of original MODIS products. Liu et al. (2020a) indicated that the classification errors in original MODIS snow products were strongly affected by omission error (OE). Our snow cover product provided a considerable improvement in this respect. The OE of our $HMRF_{solar}$-based snow products was 4.70% lower than that of original MODIS products (Table 3).



Particularly, our HMRF method is able to fill data gaps in original MODIS products in which spectral information is unavailable caused by cloud by integrating spatiotemporal and environmental information together. Table 4 shows that the OA of the $HMRF_{solar}$- and $HMRF_{ele}$- based snow cover products for those data-gap pixels in original MODIS products was

83.18% and 80.32%, which is 6.27% and 9.13% lower than that of gap-free pixels. It is also showed that the OA of $HMRF_{solar}$-based snow cover product is 2.86% higher than that of $HMRF_{ele}$-based snow cover product. Both Table 3 and Table 4 confirmed that using solar radiation as the environmental information in the HMRF framework can develop more reliable snow cover products.

**4.3 Accuracy improvement in different times of a snow season**

Previous studies (Huang et al., 2018; Klein, 2003; Li et al., 2019a) have suggested that the snow cover estimation accuracy was generally low during snow transitional period (i.e., the beginning and end of a snow season). We plotted the temporal variations in the OA, OE, and CE of the $HMRF_{solar}$- and $HMRF_{ele}$-based snow cover products (Fig. 4). The accuracy was comparatively low during snow transitional period (November to December, and February to April), whereas the accuracy was higher in snow stable period (January). The trend of the CE was almost the same as that of snow cover; that is, the

months with more snow cover had a larger CE. However, the CE was generally low in all months. By comparison with the $HMRF_{ele}$-based products, the accuracy of our new products was higher in almost all months. In addition to the stable snow period, the improvement for transition period was also notable, which increased by an average of 2.69%. In transition period, with rapid temperature changes, the snow status changed rapidly, and the snow mapped by MOD10A1 (crossing at 10:30) and MYD10A1 (crossing at 13:30) were different because of the different temperatures and solar radiation. The original

MODIS snow cover products have relatively high error during snow transitional period (Li et al., 2019). After incorporating solar radiation as an environmental contextual information in HMRF, the accuracy of snow cover products has been improved effectively.





**Table 3.** Confusion matrices between HMRF$_{solar}$-based snow products, HMRF$_{ele}$-based snow products, original MODIS snow products, and snow cover mapped from Landsat-8 imagery products for gap-free pixels during 2016–2018.

| Landsat-8 imagery | HMRF$_{solar}$-based snow products | | | HMRF$_{ele}$-based snow products | | | Original MODIS snow products | | |
|---|---|---|---|---|---|---|---|---|---|
| | Snow | Non-snow | Total | Snow | Non-snow | Total | Snow | Non-snow | Total |
| Snow | 1391834 (89.46%) | 164049 (10.54%) | 1555883 | 1320466 (84.87%) | 235417 (15.13%) | 1555883 | 1318732 (84.76%) | 237151 (15.24%) | 1555883 |
| Non-snow | 77459 (4.73%) | 1561762 (95.27%) | 1639221 | 92900 (5.67%) | 1546321 (94.33%) | 1639221 | 99849 (6.09%) | 1539372 (93.91%) | 1639221 |
| Total | 1469293 | 1725811 | 3195104 | 1413366 | 1781738 | 3195104 | 1418581 | 1776523 | 3195104 |
| Overall accuracy | 92.44% | | | 89.72% | | | 89.45% | | |





**Table 4.** Confusion matrices between HMRF$_{solar}$- based snow cover products, HMRF$_{ele}$-based snow cover products, and snow cover mapped from Landsat-8 imagery products for data-gap pixels during 2016–2018.

| Landsat-8 imagery | HMRF$_{solar}$-based snow cover product | | | HMRF$_{ele}$-based snow cover product | | |
|---|---|---|---|---|---|---|
| | Snow | Non-snow | Total | Snow | Non-snow | Total |
| Snow | 193424 (79.23%) | 50693 (20.77%) | 244117 | 185993 (76.19%) | 58124 (16.31%) | 244117 |
| Non-snow | 17402 (10.82%) | 143377 (89.18%) | 160779 | 21577 (13.42%) | 139202 (86.58%) | 160779 |
| Total | 210826 | 194070 | 404896 | 207570 | 197326 | 404896 |
| Overall accuracy | 83.18% | | | 80.32% | | |

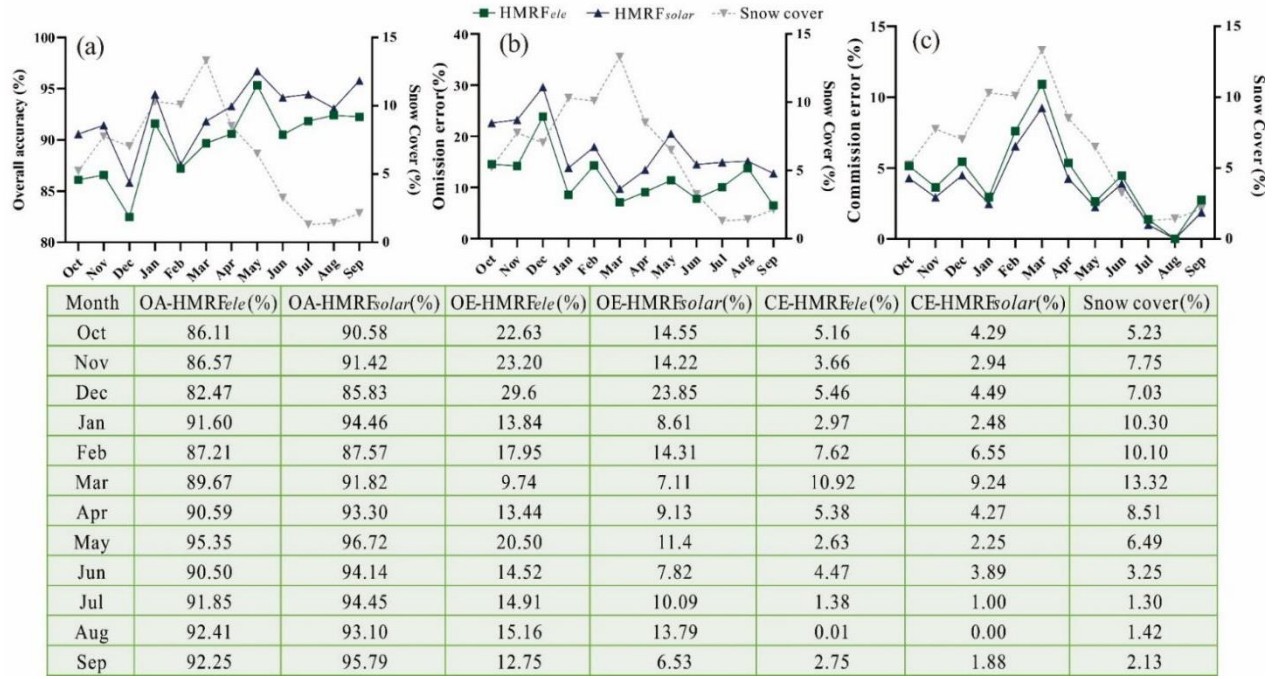

| Month | OA-HMRF$_{ele}$(%) | OA-HMRF$_{solar}$(%) | OE-HMRF$_{ele}$(%) | OE-HMRF$_{solar}$(%) | CE-HMRF$_{ele}$(%) | CE-HMRF$_{solar}$(%) | Snow cover(%) |
|---|---|---|---|---|---|---|---|
| Oct | 86.11 | 90.58 | 22.63 | 14.55 | 5.16 | 4.29 | 5.23 |
| Nov | 86.57 | 91.42 | 23.20 | 14.22 | 3.66 | 2.94 | 7.75 |
| Dec | 82.47 | 85.83 | 29.6 | 23.85 | 5.46 | 4.49 | 7.03 |
| Jan | 91.60 | 94.46 | 13.84 | 8.61 | 2.97 | 2.48 | 10.30 |
| Feb | 87.21 | 87.57 | 17.95 | 14.31 | 7.62 | 6.55 | 10.10 |
| Mar | 89.67 | 91.82 | 9.74 | 7.11 | 10.92 | 9.24 | 13.32 |
| Apr | 90.59 | 93.30 | 13.44 | 9.13 | 5.38 | 4.27 | 8.51 |
| May | 95.35 | 96.72 | 20.50 | 11.4 | 2.63 | 2.25 | 6.49 |
| Jun | 90.50 | 94.14 | 14.52 | 7.82 | 4.47 | 3.89 | 3.25 |
| Jul | 91.85 | 94.45 | 14.91 | 10.09 | 1.38 | 1.00 | 1.30 |
| Aug | 92.41 | 93.10 | 15.16 | 13.79 | 0.01 | 0.00 | 1.42 |
| Sep | 92.25 | 95.79 | 12.75 | 6.53 | 2.75 | 1.88 | 2.13 |

**Figure 4: Temporal variations in OA (a), OE (b), and CE (c) of HMRF$_{ele}$- and HMRF$_{solar}$-based snow products from 2016–2018.**

**4.4 Accuracy improvement for different surface topography**

We divided the elevation of the study area into four categories: <3000, 3000–4000, 4001–5000, and >5000 m, to explore the effect of elevation on snow cover. We then calculated the snow cover products accuracy in each category. Fig. 5 indicates that the OA of the products decreased with increasing elevation. The accuracy of our products was highest in the <3000 m category (97.79%). With increasing elevation, the OE first decreased and then increased; that is, the OE value was the lowest in the 4001–5000 m category, whereas the CE was the highest in the 4001–5000 m category. Li et al. (2020) established

effect of elevation on snow cover and also demonstrated the accuracy of original MODIS snow products generally decreased



with increasing elevation. Areas at higher elevation generally receive more solar radiation; hence, the snow status changes more rapidly at higher elevations, resulting in decreased snow cover estimation accuracy.

By comparison with the HMRF$_{ele}$-based products, the accuracy of our new products was higher in almost all elevation categories, and as the elevation increased, the accuracy improvement was more remarkable (reaching an improvement of 3.39% in the > 5000 m category). The improvements in OE first decreased and then increased; that is, the accuracy improvement of the OE value was the smallest in the 4001–5000 m category, with an improvement of 4.28%. The CE exhibited the opposite trend; that is, the accuracy improvement in the CE value was the largest in the 4001–5000 m category, at 1.2%. The accuracy improvement provided by our new snow cover product was substantially increased with the increase in elevation, indicating its effectiveness in high-altitude areas.

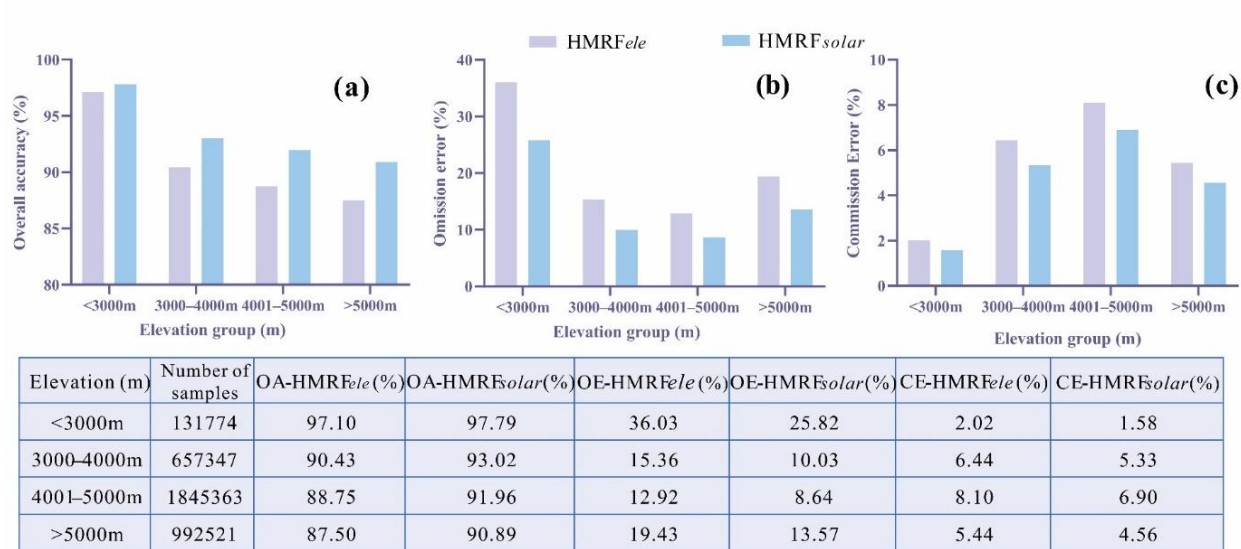

| Elevation (m) | Number of samples | OA-HMRF$_{ele}$ (%) | OA-HMRF$_{solar}$(%) | OE-HMRF$_{ele}$ (%) | OE-HMRF$_{solar}$(%) | CE-HMRF$_{ele}$ (%) | CE-HMRF$_{solar}$(%) |
|---|---|---|---|---|---|---|---|
| <3000m | 131774 | 97.10 | 97.79 | 36.03 | 25.82 | 2.02 | 1.58 |
| 3000–4000m | 657347 | 90.43 | 93.02 | 15.36 | 10.03 | 6.44 | 5.33 |
| 4001–5000m | 1845363 | 88.75 | 91.96 | 12.92 | 8.64 | 8.10 | 6.90 |
| >5000m | 992521 | 87.50 | 90.89 | 19.43 | 13.57 | 5.44 | 4.56 |

**Figure 5:** Effect of elevation on OA (a), OE (b), and CE (c) of HMRF$_{ele}$- and HMRF$_{solar}$-based snow products from 2016–2018.

To explore the effect of slope on snow cover, we divided the slope of the study area into five categories: <10°, 10°–20°, 21°–30°, 31°–40°, and >40°. Because no Landsat-8 imagery met the conditions in areas with a slope > 40°, we only calculated the accuracy of the first four categories (Fig. 6). The OA of the snow cover products decreased with increasing slope, which is consistent with the trend of the changes in accuracy with increasing elevation (Fig. 6). The accuracy of our new products was highest in the 0 –10° category, with a value of 95.91%. The UE was high for relatively low-slope areas (<30°), and the OE was high for high-slope areas (≥30°).

By comparison with the HMRF$_{ele}$-based products, the accuracy of our new products was higher in almost all slope categories, and the accuracy improvement was remarkable in the 10°–20° category (reaching an improvement of 2.47%). This means topography of 10°–20° slopes had the strongest impact on snow cover on the TP. The accuracy improvements in OE decreased with increasing slope, whereas the accuracy improvements in CE increased with increasing slope. In the 10°–20° category, the improvement in CE was the smallest, at 0.36%, whereas the improvement in OE was the largest, at 5.56%. In





the 31°–40° category, the improvement in CE was the largest, at 1.97%, whereas the improvement in OE was the smallest, at 2.04%.

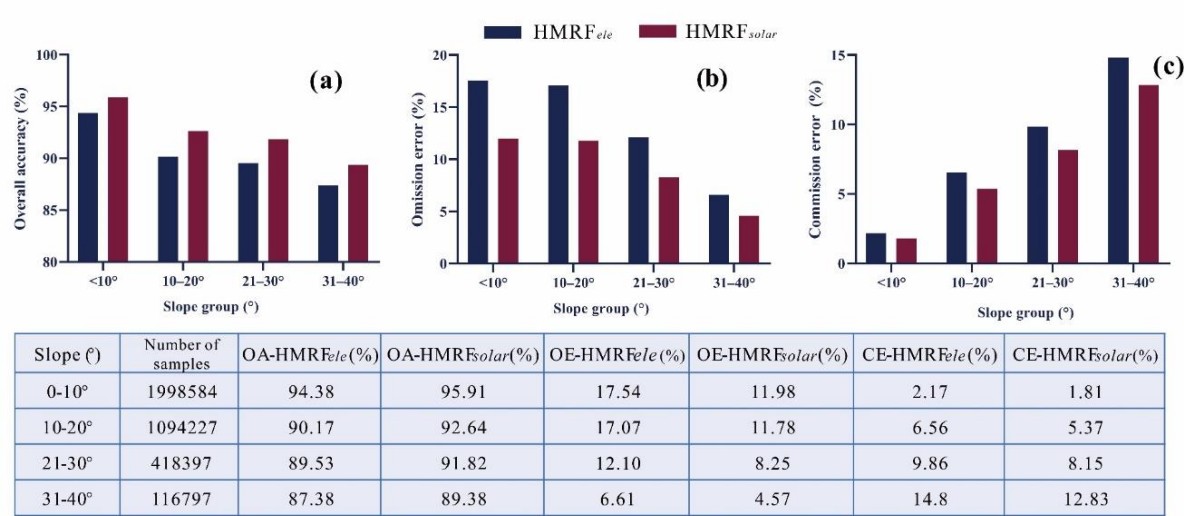

| Slope (°) | Number of samples | OA-HMRF$_{ele}$(%) | OA-HMRF$_{solar}$(%) | OE-HMRF$_{ele}$(%) | OE-HMRF$_{solar}$(%) | CE-HMRF$_{ele}$(%) | CE-HMRF$_{solar}$(%) |
|---|---|---|---|---|---|---|---|
| 0-10° | 1998584 | 94.38 | 95.91 | 17.54 | 11.98 | 2.17 | 1.81 |
| 10-20° | 1094227 | 90.17 | 92.64 | 17.07 | 11.78 | 6.56 | 5.37 |
| 21-30° | 418397 | 89.53 | 91.82 | 12.10 | 8.25 | 9.86 | 8.15 |
| 31-40° | 116797 | 87.38 | 89.38 | 6.61 | 4.57 | 14.8 | 12.83 |

340

**Figure 6: Effect of slope on OA (a), OE (b), and CE (c) of HMRF$_{ele}$- and HMRF$_{solar}$-based snow products from 2016–2018.**

Finally, we explored the effect of aspect on snow cover (Fig. 7). The OA of the snow cover products was higher on shaded slopes than on sunny slopes. The OAs of our new products on sunny and shaded slopes were 91.14% and 92.56%, respectively. By comparison with the HMRF$_{ele}$-based products, the accuracy improvement provided by our new products was remarkable on sunny slopes (improvement of 3.6%) than on shaded slopes (improvement of 2.54%), corresponding with the actual situation. The improvement in the OE on sunny slopes was also remarkable (improvement of 5.88%).

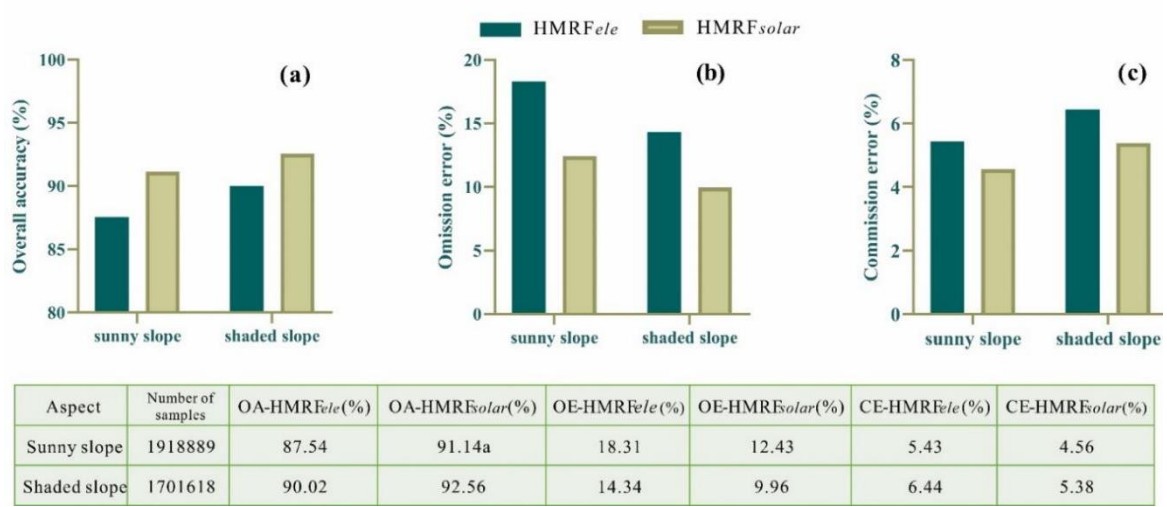

| Aspect | Number of samples | OA-HMRF$_{ele}$(%) | OA-HMRF$_{solar}$(%) | OE-HMRF$_{ele}$(%) | OE-HMRF$_{solar}$(%) | CE-HMRF$_{ele}$(%) | CE-HMRF$_{solar}$(%) |
|---|---|---|---|---|---|---|---|
| Sunny slope | 1918889 | 87.54 | 91.14a | 18.31 | 12.43 | 5.43 | 4.56 |
| Shaded slope | 1701618 | 90.02 | 92.56 | 14.34 | 9.96 | 6.44 | 5.38 |

**Figure 7: Effect of aspect on OA (a), OE (b), and CE (c) of HMRF$_{ele}$- and HMRF$_{solar}$-based snow products from 2016–2018.**

350





## 5 Discussion

### 5.1 Effect of the HMRF$_{solar}$-based snow cover products over complex terrain

Long-term and high-precision snow cover products are key to the snow hydrology research in the TP. Compared to the heterogeneous land cover, complex terrain has more severe effects on MODIS snow products in the TP (Liu et al., 2020a; Azizi and Akhtar, 2021). To enhance snow detection accuracy in mountainous areas, we used solar radiation as the environmental contextual information rather than surface elevation used in the original HMRF$_{ele}$ modelling technique (Huang et al., 2018). Fig. 8a shows a true-color Sentinel-2B image. Figs. 8b, 8c, and 8d show the original MODIS, HMRF$_{ele}$- and HMRF$_{solar}$-based snow cover products, respectively, on October 31, 2018. The examples in Fig. 8 (where the topography has a dominant effect on snow cover) show that the original HMRF$_{ele}$-based snow cover products mapped all data gaps in the shaded and sunny slopes at the same elevation as snow-covered pixels; however, after incorporating solar radiation, our new products accurately identified them as non-snow-covered pixels. In general, since sunny slopes receive relatively more solar radiation, the area of snow cover on sunny slopes is smaller than that on shaded slopes. The new HMRF$_{solar}$-based snow products more effectively filled the data gaps on sunny slopes, producing more realistic results. Solar radiation was particularly effective as an environmental contextual factor to correct the snow detection errors in mountainous areas.

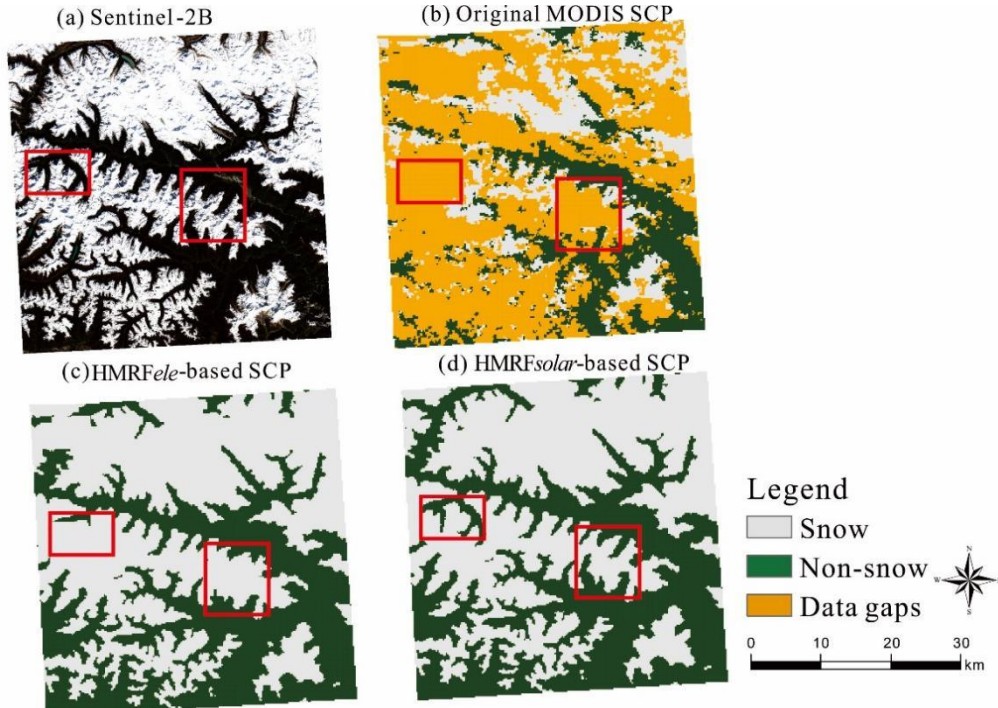

**Figure 8: Comparison between true-color Sentinel-2B imagery and(a), original MODIS snow products (b), HMRF$_{ele}$-based snow products (c) HMRF$_{solar}$-based snow products.**

## 5.2 Potential applications of the HMRF$_{solar}$-based snow cover products

370   The new snow cover products provide spatiotemporally continuous snow cover distribution at 500 m resolution over the past
two decades, thus having a great application potential for analyzing the processes of snow accumulation and melting. Figure
9 shows the snow cover trends over a sample area (location shown in Fig.1) from our daily gap-free snow products and 8-
day composite MODIS snow products on Terra (MOD10A2 v6.1) (Hall et al., 2021) during the 2017 snow season. The
MOD10A2 v6.1 products composite 8-days MODIS data to reduce the data gaps in the original daily MODIS snow products,

and represent the maximum snow cover extent in 8 days, which means once snow is identified on any day in the 8-day time
window, the pixel is mapped as snow. Although the overall variation trends of snow cover are similar between these two
products, the snow cover percentage obtained from the MOD10A2 v6.1 data was 19.70% higher than that of daily products
during 2017 snow season. The status of the snow in the TP changes rapidly. Compared with the 8-day composite MODIS
snow products (MOD10A2 v6.1), the HMRF$_{solar}$-based snow cover products can get more detailed and accurate information

on snow accumulation and melting processes, particularly during the snow transitional periods (February to April, and
November to December) when short-term snowfall events occur frequently (Fig.9). In addition, owing to its high temporal
resolution (daily) and long-term spatiotemporal continuity, the generated snow cover products also provide important
baseline information for monitoring climate change, calibrating hydrological models, and simulating snowmelt runoff in the
TP

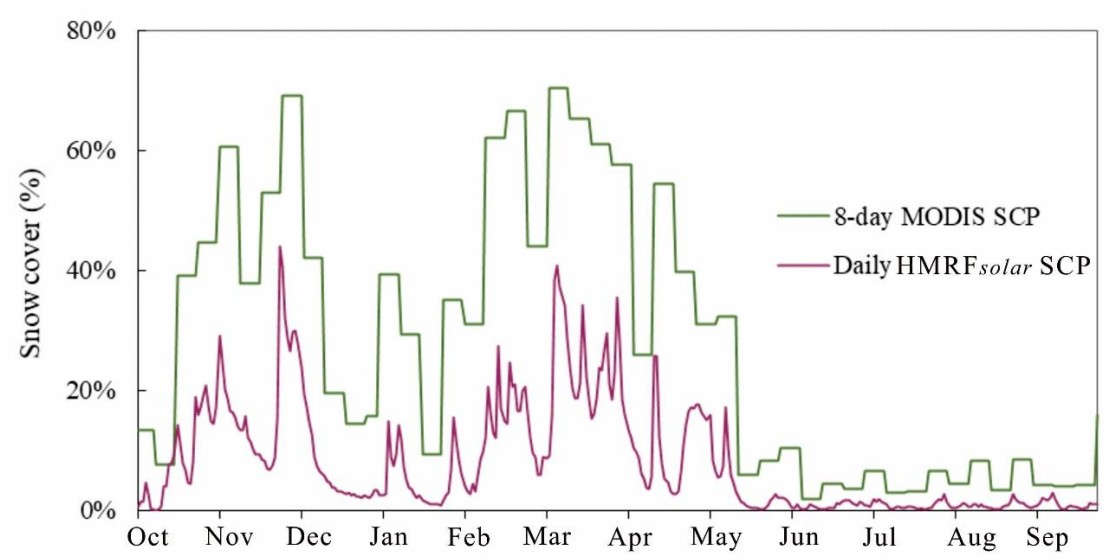


**Figure 9: Snow cover percentage for daily HMRF$_{solar}$-based snow cover products and 8-day composite MODIS snow products in a sample area during the 2017 snow season of the TP. (SCP stands for snow cover products.)**



## 5.3 Limitations of the HMRF$_{solar}$-based snow cover products

The accuracy evaluation for the new snow cover products had some limitations. The majority of *in situ* observations in the
TP are distributed in the low-altitude areas in the east, with only 22% located in high-altitude areas (Fig. 1a). The number of
*in situ* observations in the snow category was considerably smaller than that in the non-snow category, which created
challenges in verifying snow category classification (Zhang et al., 2021). Liu et al. (2020a) demonstrated that compared to *in
situ* observations, snow cover mapped from Landsat series images is more related to MODIS snow cover products owing to
the closer spatial match. Therefore, the conclusions about the new snow products were based on the accuracy evaluation
against Landsat-8 imagery, which may still contain some bias. Additionally, the solar radiation indicated used only imitates
the solar radiation under clear-sky conditions. In a follow-up study, we will simulate the solar radiation throughout the year
according to the weather conditions (e.g. the effect of cloud cover) and use parallel calculations to enhance calculation
efficiency and further improve the snow cover product accuracy.

## 6 Data availability

The long-term daily snow cover products produced here are gap-free at 500 m resolution under the Universal Transverse
Mercator (UTM zone 45) projection, and can be freely accessed from the National Tibetan Plateau Data Center at
https://doi.org/10.11888/Cryos.tpdc.272204 (Huang and Xu, 2022), which is stored as a zip file (∼ 1.56 GB) for each year.
By uncompressing the zip file, the daily snow cover data is provided in GeoTIFF format, and the values in the snow cover
products are classified as snow (1) and non-snow (2). The name of each file is "HMRFSTP_yyyyddd.tif", in which
HMRFSTP is the abbreviation of "Hidden Markov Radom Field -based snow cover products for Tibetan Plateau", yyyy
stands for year and DDD stands for Julian day, such as 2002135.tif describes the snow cover on Tibet Plateau on the 135th
day of 2002.

## 7 Conclusions

In this study, we generated long-term daily gap-free snow cover products at 500 m resolution from original MODIS snow
products in the TP over the past two decades. The snow cover estimate accuracy was greatly improved by incorporating solar
radiation as a surrogate for environmental contextual information in HMRF framework in mountainous areas. We validated
and evaluated our snow cover products through comparison with *in situ* observations and high-resolution snow cover
mapped from Landsat-8 imagery, with accuracy estimate of 98.31% and 92.44%, respectively. Our evaluation also suggests
that incorporating solar radiation, instead of the simple use of surface elevation as the environmental contextual information
in HMRF framework, results in the accuracy of the snow products increase by 2.72% and the omission error decrease by
4.59%. Specifically, the accuracy of the new snow products is particularly improved during snow transitional period and
over complex terrains with high elevation as well as sunny slopes.

We believe that the long-term and spatiotemporally continuous snow cover products generated in this study have great potential to analyse the processes of snow accumulation and melting, to monitor the climate change, and to understand the
hydrological cycling in the TP.

**Author contributions.**

HY and LHX developed the HMRF methodology and designed the study. XJH and ZYL applied the methodology and performed the validation. XJY and XWJ performed the visualization. YBL, WSJ, and WJP analyzed the results. ZZJ worked on data curation. Both the authors devoted to the writing of the paper and data quality control.

**Competing interests.**

The authors declare that they have no conflict of interest.

**Disclaimer.**

The views and interpretations in this publication are those of the authors and are not necessarily attributable to the ICI-MOD.

**Special issue statement.**

This article is part of the special issue "Extreme environment datasets for the three poles". It is not associated with a conference.

**Acknowledgments.**

We would like to thank the National Meteorological Centre of China for providing the *in situ* observations over the Tibetan Plateau.

**Financial support.**

This work was supported by the National Natural Science Foundation of China (no. 42125604, 42071306 and 41701502).

**Review statement.**

This paper was edited by XX and reviewed by two anonymous referees.





## Appendix

Table A1. Landsat-8 OLI images used for assessment of the HMRF-based snow cover products in this study.

| Image pair No. | OLI tile path/row | Date of acquisition | Imagery UTC | Cloud cover (%) |
|---|---|---|---|---|
| 1 | 131/37 | 5-Jan-16 | 3:44:45 | 3.54 |
| 2 | 131/37 | 16-Apr-18 | 3:43:59 | 0.82 |
| 3 | 131/39 | 25-Mar-16 | 3:45:13 | 1.38 |
| 4 | 132/34 | 17-Apr-16 | 3:49:14 | 2.70 |
| 5 | 133/37 | 29-Mar-18 | 3:56:31 | 1.27 |
| 6 | 133/38 | 3-Jan-16 | 3:57:32 | 3.17 |
| 7 | 133/40 | 4-Dec-16 | 3:58:33 | 4.72 |
| 8 | 134/35 | 26-Jan-16 | 4:02:30 | 4.51 |
| 9 | 134/37 | 27-Dec-16 | 4:03:27 | 3.58 |
| 10 | 134/37 | 27-Oct-17 | 4:03:31 | 1.65 |
| 11 | 134/38 | 27-Feb-16 | 4:03:31 | 1.60 |
| 12 | 134/39 | 18-Apr-17 | 4:03:27 | 2.70 |
| 13 | 134/40 | 16-Feb-18 | 4:04:13 | 2.24 |
| 14 | 134/40 | 20-Mar-18 | 4:03:57 | 2.37 |
| 15 | 135/33 | 11-May-17 | 4:07:11 | 3.17 |
| 16 | 135/36 | 19-Nov-17 | 4:09:14 | 4.05 |
| 17 | 135/38 | 14-May-18 | 4:08:50 | 4.12 |
| 18 | 135/39 | 5-Mar-16 | 4:10:05 | 2.53 |
| 19 | 135/39 | 4-Feb-17 | 4:10:13 | 1.52 |
| 20 | 135/39 | 21-Dec-17 | 4:10:24 | 3.47 |
| 21 | 136/33 | 5-Jul-17 | 4:13:47 | 2.89 |
| 22 | 136/37 | 22-Oct-16 | 4:15:56 | 2.71 |
| 23 | 136/38 | 6-Jun-18 | 4:14:45 | 3.44 |
| 24 | 137/39 | 29-Oct-16 | 4:22:55 | 2.75 |
| 25 | 137/39 | 22-Mar-17 | 4:22:12 | 4.54 |
| 26 | 138/37 | 16-May-17 | 4:27:23 | 2.79 |
| 27 | 138/38 | 8-Jan-17 | 4:28:32 | 4.28 |
| 28 | 139/35 | 17-Dec-17 | 4:33:31 | 0.89 |
| 29 | 140/35 | 14-Jul-16 | 4:39:27 | 0.18 |
| 30 | 140/35 | 2-Oct-16 | 4:39:45 | 1.71 |
| 31 | 140/36 | 2-Aug-17 | 4:39:52 | 0.15 |
| 32 | 140/41 | 8-Dec-17 | 4:42:02 | 4.45 |
| 33 | 141/34 | 3-Jun-16 | 4:44:57 | 3.62 |
| 34 | 141/35 | 10-Sep-17 | 4:45:48 | 1.29 |
| 35 | 141/40 | 14-Feb-17 | 4:47:37 | 3.12 |

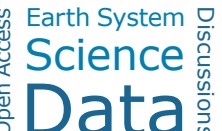

| 36 | 141/40 | 16-Nov-18 | 4:47:37 | 2.03 |
| 37 | 142/39 | 9-May-16 | 4:53:02 | 9.35 |
| 38 | 142/39 | 4-Jan-17 | 4:53:40 | 2.72 |
| 39 | 143/39 | 5-Sep-16 | 4:59:51 | 2.12 |
| 40 | 143/39 | 4-Jun-17 | 4:59:16 | 4.68 |
| 41 | 144/35 | 23-May-16 | 5:03:50 | 3.79 |
| 42 | 144/35 | 14-Oct-16 | 5:04:33 | 3.08 |
| 43 | 144/35 | 29-Jul-17 | 5:04:10 | 0.84 |
| 44 | 144/39 | 21-Apr-16 | 5:05:21 | 6.73 |
| 45 | 145/35 | 22-Sep-17 | 5:10:35 | 3.93 |
| 46 | 145/39 | 22-Nov-16 | 5:12:21 | 1.57 |
| 47 | 146/35 | 14-Jul-18 | 5:15:45 | 2.16 |
| 48 | 146/36 | 25-Jun-17 | 5:16:44 | 4.44 |
| 49 | 146/38 | 4-Feb-18 | 5:17:39 | 3.27 |
| 50 | 147/35 | 7-Nov-17 | 5:23:03 | 1.66 |
| 51 | 147/36 | 21-Jan-16 | 5:23:15 | 1.81 |
| 52 | 147/36 | 9-Mar-16 | 5:23:02 | 4.26 |
| 53 | 147/36 | 20-Sep-17 | 5:23:19 | 1.93 |
| 54 | 148/35 | 29-Oct-17 | 5:29:15 | 1.97 |
| 55 | 149/34 | 1-Aug-17 | 5:34:42 | 4.71 |
| 56 | 149/35 | 2-Nov-16 | 5:35:29 | 2.72 |
| 57 | 149/35 | 20-Dec-16 | 5:35:22 | 3.92 |
| 58 | 149/35 | 11-Apr-17 | 5:34:37 | 1.08 |
| 59 | 149/35 | 4-Aug-18 | 5:34:27 | 2.57 |
| 60 | 150/33 | 27-Feb-16 | 5:40:25 | 3.72 |
| 61 | 150/33 | 18-Jun-16 | 5:40:14 | 2.06 |
| 62 | 150/33 | 20-May-17 | 5:40:00 | 3.55 |

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
