# Peer review of "HMRFS-TP: long-term daily gap-free snow cover products over the Tibetan Plateau from 2002 to 2021 based on Hidden Markov Random Field model"

_Earth System Science Data, 2022_

## Author Comment (AC1)

**Response to Reviewer #1**

**Comments to the Author:**

This paper generated long-term daily gap-free snow cover products in the Tibetan Plateau over the past two decades by optimally integrating spectral, spatiotemporal, and environmental information within a Hidden Markov Radom Field model. From the report of this paper, the accuracy of the new snow cover products was greatly improved during the snow transitional period and over complex terrains as well as sunny slopes. As a spatiotemporally continuous and high-quality snow cover product is essential for cryospheric science, the produced long time-series daily snow cover product could be a significant dataset for understanding climate change and the water cycle over the Tibetan Plateau. The paper is scientifically sounding. Despite its significance, several issues still need to be resolved before a publication to ESSD. The introduction about the GPU-accelerated model, and why you chose this sample area to illustrate the snow cover percentage obtained from your daily snow cover products and MODIS 8-day composite products could be sufficiently explained. Besides, the total accuracy, omission error, and commission error of your new snow cover products should be compared and discussed with the accuracy reported by other studies. In addition, some figures need to be revised.

**Response:** We truly appreciate the constructive suggestions and comments. We have revised our manuscript to our best effort. In this revision, we have refitted the empirical relationship between snow fraction and NDSI, and reprocessed the input data for HMRF modeling. We have regenerated a more rigorous daily gap-free snow cover dataset. In addition, we have added longer time series and terrain-corrected Landsat images for validation, including Landsat-5 TM, Landsat-7 ETM+, and Landsat-8 OLI images. The new accuracy assessment demonstrates the effect and potential applications of our new daily snow cover dataset.

**Comment 1:**

L40, which rivers are there, please given specific examples.

**Response:** We have modified the sentence to "…the runoff of numerous rivers such as the Yangtze and Yellow rivers (Immerzeel et al., 2010)" (now in Line 40-41).

**Comment 2:**

L75, the abbreviation of HMRF is already defined on L66 and does not need to be defined again.

**Response:** We have removed the redundant definition (now in Line 76).

**Comment 3:**

L91, the numbers here appear to be inserted as formulas.

**Response:** We have modified the number format (now in Line 92).

**Comment 4:**

L91-95, the importance of the Tibetan Plateau as a water tower in Asia and the importance of snow in it should be highlighted.

**Response:** As suggested, we have added the following sentences in Line 95-99:

*Seasonal snow cover on the TP has a great potential to influence the hydrological cycle and heat wave frequency in northern China (Wu et al., 2012). In addition, seasonal snow accumulation on the TP is an important part of surface water accumulation in southwestern China and surrounding countries. Several major rivers in China and surrounding Asian countries, such as the Yangtze, Yellow, Mekong, Salween, Brahmaputra, Ganges, and Indus Rivers, all originate from the TP.*

*Reference:*

Wu, Z., Jiang, Z., Li, J., Zhong, S. and Wang, L.: Possible association of the western Tibetan plateau snow cover with the decadal to interdecadal variations of northern China heatwave frequency, Climate Dynamics, 39(9–10),2393–2402, 2012.

**Comment 5:**

Figure 1, change the frame color of the sample area, as its color is very close to the color of the stations.

**Response:** As suggested, we have modified our figure 1 as following:

[Figure]

Figure 1: Topography, meteorological stations, and survey photos of the TP. (a) Surface elevation and distribution of meteorological stations in the TP. Landsat series data utilized for

validation and sample area are also shown. (b) and (c) in situ photos from field survey in the TP.

**Comment 6:**
In Figure 3, some text overlaps with the frame, please check and revise.
**Response:** As suggested, we have corrected the text in Figure 3.

[Figure]

Figure 3: Overall flowchart of the $HMRF_{solar}$-based framework. (SCP stands for snow cover products)

**Comment 7:**
L192, change "snow product is" to "snow products are".
**Response:** we have changed the text as suggested.

**Comment 8:**
Line 222, the authors stated that they used a GPU-accelerated model. It is suggested to provide more detailed information about the GPU configuration.
**Response:** We have provided more detailed information about the GPU configuration in Line 237-239:

*A general-purpose desktop computer was used to test the parallel computational efficiency. The computer has an Intel Core™ i7-10700 CPU (16 cores and max clock rate is at 2.90 GHz), an NVIDIA GeForce RTX 2070 SUPER Card with 2560 cores and 16240 MB global memory, and Windows 11 Ultimate 64-bit Operation System.*

**Comment 9:**
L255-290, can the authors compare the overall accuracy, omission error, and commission error of new snow cover products with the accuracy reported by other studies?

**Response:** In this revision, we have compared the overall accuracy, omission error, and commission error of new snow cover products with the accuracy reported by previous studies on Line 414-417:

*Compared with in situ observations, the overall accuracy of snow cover products on the TP reported by other studies is in the range of 90.74%-96.6%, and the omission error is greater than the commission error (Yu et al., 2016; Qiu et al., 2017; Xu et al., 2017; Zheng and Cao, 2019). The overall accuracy of our new snow products is 98.29% in comparison with in situ observations, and the new product has a considerable improvement in omission error.*

*References:*

Qiu, Y., Zhang, H., Chu, d., and Xuecheng, Z.: Cloud removing algorithm for the daily cloud free MODIS-based snow cover product over the Tibetan Plateau, Journal of Glaciology and Geocryology, 39(3), 515-526, doi: 10.7522/j.issn.1000-0240.2017.0058, 2017.

Xu, W. F., Ma, H. Q., Wu, D. H., and Yuan, W. P.: Assessment of the daily cloud-free MODIS snow-cover product for monitoring the snow-cover phenology over the Qinghai-Tibetan Plateau, Remote Sensing, 9, ARTN 585,doi: 10.3390/rs9060585, 2017.

Yu, J., Zhang, G., Yao, T., Xie, H., Zhang, H., Ke, C., and Yao, R.: Developing daily cloud-free snow composite products from MODIS Terra–Aqua and IMS for the Tibetan Plateau, IEEE Transactions on Geoscience and Remote Sensing, 54, 2171-2180, doi: 10.1109/tgrs.2015.2496950, 2016.

Zheng, Z. and Cao, G.: Snow cover dataset based on multi-source remote sensing products blended with 1km spatial resolution on the Qinghai-Tibet Plateau (1995-2018), National Tibetan Plateau Data Center, doi: 10.11888/Snow.tpdc.270102, 2019.

**Comment 10:**

L291, please add the definition of "snow season".

**Response:** The definition of "snow season" has been added in Line 317-318:

*To explore the details of snow cover variation, we define the snow season from September of previous year to August of the following year, for example, the time range of the 2002 snow season is 2002.9.1-2003.8.31 (Chen et al., 2018b).*

*Reference:*

Chen, X., Long, D., Hong, Y., Hao, X., and Hou, A.: Climatology of snow phenology over the Tibetan plateau for the period 2001-2014 using multisource data, International Journal of Climatology, 38, 2718-2729, doi: 10.1002/joc.5455, 2018b.

**Comment 11:**

L295-296, it is suggested to use the threshold of 90% of overall accuracy to summarize the status of monthly accuracy.

**Response:** In this revision, we have used the threshold of 90% of overall accuracy to summarize the monthly accuracy in Line 321-324:

*Except November, December, and February, the OAs of the new snow products were more than 90% in all months. The accuracy was relatively low during snow*

*transitional period (November, December, and February to April), whereas the accuracy was higher in snow stable period.*

**Comment 12:**
In figure 4, figure 5, figure 6, and figure 7, you should provide the improvements of OA, OE, and CE of your new snow products.

**Response:** We have added the improvements of OA, OE, and CE of the new snow products in figure 4, figure 5, figure 6, and figure 7.

[Figure]

| Month | OA-HMRF*ele* | OA-HMRF*solar* | OE-HMRF*ele* | OE-HMRF*solar* | CE-HMRF*ele* | CE-HMRF*solar* | Snow cover |
|---|---|---|---|---|---|---|---|
| Sep | 87.62% | 90.02% (+2.40%) | 19.21% | 13.92% (-5.29%) | 4.37% | 3.09% (-1.28%) | 5.23% |
| Oct | 88.61% | 91.87% (+3.26%) | 20.76% | 14.53% (-6.23%) | 6.13% | 4.98% (-1.15%) | 7.75% |
| Nov | 87.26% | 90.03% (+1.91%) | 21.59% | 16.01% (-5.58%) | 4.09% | 3.19% (-0.90%) | 7.03% |
| Dec | 82.14% | 86.72% (+4.58%) | 21.84% | 18.37% (-3.47%) | 6.98% | 5.27% (-1.71%) | 10.30% |
| Jan | 90.29% | 92.17% (+1.88%) | 13.11% | 10.36% (-2.75%) | 6.15% | 3.59% (-2.56%) | 10.10% |
| Feb | 82.24% | 84.22% (+1.98%) | 25.49% | 21.29% (-4.20%) | 10.23% | 8.94% (-1.29%) | 13.32% |
| Mar | 89.03% | 90.78% (+1.75%) | 14.17% | 10.49% (-3.68%) | 12.82% | 10.17% (-2.65%) | 8.51% |
| Apr | 92.64% | 93.22% (+0.70%) | 17.64% | 14.72% (-2.92%) | 11.31% | 9.46% (-1.85%) | 6.49% |
| May | 92.32% | 94.40% (+2.08%) | 15.76% | 13.36% (-2.40%) | 10.31% | 8.29% (-2.02%) | 3.25% |
| Jun | 92.37% | 93.86% (+1.22%) | 12.38% | 10.39% (-1.99%) | 8.46% | 7.21% (-1.25%) | 1.30% |
| Jul | 90.53% | 91.07% (+2.69%) | 10.01% | 8.31% (-1.70%) | 2.33% | 2.14% (-0.19%) | 1.42% |
| Aug | 88.86% | 90.73% (+1.87%) | 14.15% | 12.82% (-1.33%) | 2.16% | 1.88% (-0.28%) | 2.13% |

Figure 4: Temporal variations in OA (a), OE (b), and CE (c) of HMRF*ele*- and HMRF*solar*-based snow products from 2002–2021.

[Figure]

| Elevation (m) | OA-HMRF*ele* | OA-HMRF*solar* | OE-HMRF*ele* | OE-HMRF*solar* | CE-HMRF*ele* | CE-HMRF*solar* |
|---|---|---|---|---|---|---|
| <3000m | 92.96% | 94.26% (+1.30%) | 27.31% | 20.98% (-6.33%) | 2.16% | 1.98% (-0.18%) |
| 3000–4000m | 89.15% | 91.33% (+2.18%) | 19.84% | 14.69% (-5.15%) | 7.23% | 5.91% (-1.32%) |
| 4001–5000m | 87.12% | 89.60% (+2.48%) | 13.24% | 9.77% (-3.47%) | 9.12% | 6.79% (-2.33%) |
| >5000m | 85.09% | 88.25% (+3.16%) | 20.37% | 16.12% (-4.25%) | 6.36% | 5.16% (-1.20%) |

Figure 5: Effect of elevation on OA (a), OE (b), and CE (c) of HMRF*ele*- and HMRF*solar*-based snow products from 2002–2021.

[Figure]

Figure 6: Effect of slope on OA (a), OE (b), and CE (c) of $HMRF_{ele}$- and $HMRF_{solar}$-based snow products from 2002–2021.

[Figure]

Figure 7: Effect of aspect on OA (a), OE (b), and CE (c) of $HMRF_{ele}$- and $HMRF_{solar}$-based snow products from 2002–2021.

**Comment 13:**

Figure 7, change "91.14a" to "91.14".

**Response:** "91.14a" has been changed to "91.14" in figure 7.

**Comment 14:**

L375, explain why you chose this sample area to illustrate the snow cover percentage obtained from your daily snow cover products and MODIS 8-day composite products?

**Response:** We chose this sample area because it covers more mountainous areas and has seasonal snow for mutiple years, which makes it a perfect site to detect snow accumulation and melting. We have added more detials in Line 421-423.

---

## Author Comment (AC2)

**Response to Reviewer #2**

**Comments to the Author:**

Snow cover plays an essential role in climate change and the hydrological cycle of the Tibetan Plateau. Currently optical sensors are severely affected by clouds, resulting in a gap in snow products. Using MODIS snow cover product and HMRF algorithm, this work produced daily cloud-free snow cover dataset from 2002 to 2021 over the Tibetan Plateau. In order to validate the accuracy of the dataset, the authors used snow depth data from ground meteorological stations and Landsat-8 images as reference data to systematically evaluate the accuracy of snow products produced from different altitudes and slopes. And this work improved the elevation representing environmental information of the original HMRF model with solar radiation based on the experience of actual field experiments, and the validation results showed its great effect on the accuracy improvement. However, there are still some issues needed to be justified clearly.

**Response**: We are very grateful for the constructive suggestions and comments from the reviewer. We have significantly improved the manuscript in this revision. In this revision, we have refitted the empirical relationship between snow fraction and NDSI, and reprocessed the input data for HMRF modeling. We have regenerated a more rigorous daily gap-free snow cover dataset. In addition, we have added longer time series and terrain-corrected Landsat images for validation, including Landsat-5 TM, Landsat-7 ETM+, and Landsat-8 OLI images. The new accuracy assessment demonstrates the effect and potential applications of our new daily snow cover dataset. Please see our responses below.

The threshold of NDSI used in this work is set as 0.4, while in the work of Zhang et al., (2020), they used the value of NDSI as 0.1 to determine snow or not in the Tibetan Plateau. So, I'd like suggest the authors have to compare these two threshold on the determination of snow cover in Tibetan Plateau.

**Reference:** Zhang, H., Zhang, F., Che, T., & Wang, S. (2020). Comparative evaluation of VIIRS daily snow cover product with MODIS for snow detection in China based on ground observations. Science of The Total Environment, 724, 138156.

**Response:** We appreciate the reference and suggestion from the reviewer. In this revision, we first compared the extracted results by using the threshold of NDSI as 0.1 and 0.4 with *in situ* observation (Table R1). The overall accuracy of NDSI with the threshold of 0.4 (97.39%) was higher than that of NDSI with the threshold of 0.1 (95.24%). However, we also found that the accuracy of snow category with NDSI threshold of 0.4 (68.37%) was lower than that with NDSI threshold of 0.1 (83.11%). The reason for the high overall accuracy with NDSI threshold of 0.4 is that the accuracy of non-snow category with this threshold is higher, and the large non-snow samples (89 times the snow category) enhanced the overall accuracy. Due to the small number of snow category samples using *in situ* observation (only 882), we further selected Landsat series data with similar amount of snow and non-snow samples for further verification (Table R2). In this case, the overall accuracy with NDSI threshold

of 0.4 (83.50%) was still higher than that with NDSI threshold of 0.1 (77.28%). When the threshold is set as 0.4, although the accuracy of snow category (82.64%) is lower than that of 0.1 threshold (94.93%), the accuracy for non-snow category (84.36%) is much higher than that of 0.1 threshold (59.63%). In our experiments, using 0.1 as the NDSI threshold in the Tibetan Plateau may cause too many non-snow pixels to be misclassified as snow pixels.

We further explored why our finding is different from Zhang et al. (2020)' results. The study area of Zhang et al. (2020) is the entire China, while ours is the Tibetan Plateau. First, the thresholds of the measured *in situ* snow depth data used for validation are different. The *in situ* snow depth used for verification in Zhang et al., (2020) is divided by a 1 cm threshold. Ke et al. (2016) demonstrates that thin snow depth reduces the reliability of snow-related studies in China. In our study, a 3 cm threshold was utilized to classify the *in situ* snow depth (Huang et al., 2022). Different snow depth thresholds lead to various snow classifications, which lead to different results. In addition, the numbers of snow and non-snow samples used for validation are also different between our two studies. We selected the Landsat series data with similar amount of snow and non-snow samples for verification, while Zhang et al., (2020) used the *in situ* snow depth with more non-snow samples.

We also compared our results with other snow studies over the Tibetan Plateau. Gao et al. (2019) explored the optimal NDSI threshold for snow cover identification on the Tibetan Plateau under different land cover types, and verified the accuracy with Landsat-5 TM and Landsat-8 OLI data. Their results show that the optimal NDSI thresholds are 0.33, 0.40, and 0.47 under grassland, sparse vegetation surface types, and other underlying surface types, respectively. Since our study did not divide the Tibetan Plateau into different land cover types, a threshold of 0.4 was selected based on our experimental results and as referenced to existing literature. The suggestion of the reviewer has given us a good inspiration. In our future research, we will explore other optimal threshold of NDSI for snow identification in the Tibetan Plateau.

**Table R1** Confusion matrices between MODIS snow products with different threshold of NDSI, and *in situ* observation during 2002–2021.

| *In situ* observation | NDSI with a threshold of 0.4 | | | NDSI with a threshold of 0.1 | | |
|---|---|---|---|---|---|---|
| | Snow | Non-snow | Total | Snow | Non-snow | Total |
| Snow | 603 (68.37%) | 279 (31.63%) | 882 | 733 (83.11%) | 149 (16.89%) | 882 |
| Non-snow | 1789 (2.28%) | 76696 (97.72%) | 78485 | 3629 (4.62%) | 74856 (95.38%) | 78485 |
| Total | 2392 | 76975 | 79367 | 4362 | 75005 | 79367 |
| Overall accuracy | 97.39% | | | 95.24% | | |

**Table R2** Confusion matrices between MODIS snow products with different threshold of NDSI, and snow cover mapped from Landsat series observation during 2002–2021.

| Landsat series | NDSI with a threshold of 0.4 | | | NDSI with a threshold of 0.1 | | |
|---|---|---|---|---|---|---|
| | Snow | Non-snow | Total | Snow | Non-snow | Total |
| Snow | 239056 (82.64%) | 50227 (17.36%) | 289283 | 274602 (94.93%) | 14681 (5.07%) | 289283 |

| | 45235 (15.64%) | 244048 (84.36%) | 289283 | 116790 (40.37%) | 172493 (59.63%) | 289283 |
|---|---|---|---|---|---|---|
| Non-snow | 45235 (15.64%) | 244048 (84.36%) | 289283 | 116790 (40.37%) | 172493 (59.63%) | 289283 |
| Total | 284291 | 294275 | 578566 | 391392 | 187174 | 578566 |
| Overall accuracy | **83.50%** | | | **77.28%** | | |

*References:*

Gao, Y., Hao, X. H., He, D. C.,Huang, G. H., Wang, J., Zhao, H. Y., Wei., Y. R., Shao., D. H., Wang., W. G.: Snow cover mapping algorithm in the Tibetan Plateau based on NDSI threshold optimization of different land cover types, Journal of Glaciology and Geocryology, 41(5), 1162-1172, doi: 10.7522/j.issn.1000-0240.2019.1155, 2019. (in Chinese)

Huang, Y., Song, Z. C., Yang, H. X., Yu, B. L., Liu, H. X., Che, T., Chen, J., Wu, J. P., Shu, S., Peng, X. B., Zheng, Z. J., and Xu, J. H.: Snow cover detection in mid-latitude mountainous and polar regions using nighttime light data, Remote Sensing of Environment, 268, doi: 10.1016/j.rse.2021.112766, 2022.

Ke, C. Q., Li, X. C., Xie, H. J., Ma, D. H., Liu, X., Cheng, K.,: Variability in snow cover phenology in China from 1952 to 2010, Hydrology and Earth System Sciences, 20, 755, doi: 10.5194/hess-20-755-2016, 2016.

The Tibetan Plateau has high altitude and complex terrain, and Landsat-8 data used for reference data is 30 m, which will be affected by terrain and mountain shadow. Have you considered the terrain effect on Landsat-8 snow cover? And how's it affect validation results?

**Response:** We agree that Landsat data may be influenced by topography and mountain shadow to some extent, which can result in underestimation of snow cover in the Tibetan Plateau. In this revision, we applied a classic topographic correction model, C correction model (Teilet et al., 1982), to correct for the terrain effect on all Landsat series data used in this study (now in Line 151-152).

*Reference:*

Teilet, P. M., Guindon, B., Goodenough, D. G.: On the slope-aspect correction of multispectral scanner data, Canadian Journal of Remote Sensing, 8(2), 1537-1540, doi: 10.1080/07038992.1982.10855028, 1982.

Why use solar radiation not net radiation to represent environmental effect? Net radiation might be more related with snow surface than solar radiation here.

**Response:** Net radiation is defined as the difference between incoming and outgoing radiation flux. Solar radiation we used in this work applies latitude, slope, aspect, date, and interval time as inputs, and estimates direct, diffuse, reflected solar energy received by the ground. The complex topography of the Tibetan Plateau determines the availability of radiation at specific locations. Compared with the net radiation, solar radiation takes into account the effect of terrain (i.e., latitude, slope, aspect) and seasons (i.e., date) more comprehensively, which is very necessary for the Tibetan Plateau with complex terrain conditions. So, we used solar radiation to represent environmental effect.

The snow fraction estimated method used in the equation (2) was derived through other regions, and many studies have shown that the linear relationship has limited accuracy in the Tibetan Plateau region. If possible, I'd like suggest the authors re-fit that empirical relationship between snow fraction and NDSI in the Tibetan Plateau region. In addition, the fitting relations of Terra and Aqua satellites are different. If the same equation was used for Terra and Aqua, it might cause error on snow cover determination.

**Response:** According to the reviewer's comments, we have re-fitted the empirical relationship between snow fraction and NDSI of Terra and Aqua in the Tibetan Plateau using Landsat series data over 20 years (Eq.1 and Eq.2):

$$P(x_i|\beta_1)_{Terra} = (1.222 \times NDSI + 0.038)/100 \qquad (1)$$
$$P(x_i|\beta_1)_{Aqua} = (1.164 \times NDSI + 0.058)/100 \qquad (2)$$

The sample points used for re-fit Terra and Aqua were 972884, and 952221, respectively, and the correlation coefficients of the empirical relationship of Terra and Aqua satellites were 0.86 and 0.89, respectively (now in Line 192-199). Due to the re-fit of the FSC based on Eq.1 and Eq.2, we also recalculated the optimal parameters and reproduced the dataset. The new calculated optimal parameters of the HMRF*solar* model for spectral, spatial-temporal, and environmental information of the TP were 0.117, 1.294, and 0.532, respectively (now in Line 397-403). The overall accuracy of the reproduced dataset was 91.36%, which increased by 2.06% compared with the overall accuracy of original MODIS products (Table 3, in Sections 4.1 and 4.2).

**Table 3.** Confusion matrices between HMRF$_{solar}$-based snow products, HMRF$_{ele}$-based snow products, original MODIS snow products, and snow cover mapped from Landsat series data products for gap-free pixels during 2002–2021.

| Landsat series data | HMRF$_{solar}$-based snow products | | | HMRF$_{ele}$-based snow products | | | Original MODIS snow products | | |
|---|---|---|---|---|---|---|---|---|---|
| | Snow | Non-snow | Total | Snow | Non-snow | Total | Snow | Non-snow | Total |
| Snow | 916593 (85.06%) | 160936 (14.94%) | 1077529 | 901202 (83.64%) | 176327 (16.36%) | 1077529 | 881646 (81.82%) | 195883 (18.18%) | 1077529 |
| Non-snow | 108214 (5.31%) | 1931065 (94.69%) | 2039279 | 120590 (5.91%) | 1918689 (94.09%) | 2039279 | 137391 (6.74%) | 1901888 (93.26%) | 2039279 |
| Total | 1024807 | 2092001 | 3116808 | 1021792 | 2095016 | 3116808 | 1019037 | 2097771 | 3116808 |
| Overall accuracy | 91.36% | | | 90.47% | | | 89.31% | | |

Landsat-8 images was not enough to demonstrate the current results. If possible, please add more validation Landsat images, such as Landsat-5/7 images.

**Response:** In this revision, we have added more Landsat images for validation, including Landsat-5 TM, Landsat-7 ETM+, and Landsat-8 OLI images. The detailed information of the Landsat images is shown in Table A1.

Table A1. Landsat series images used for assessment of the HMRF-based snow cover products in this study.

| Image pair No. | Sensor | Tile path/row | Date of acquisition | Cloud cover (%) |
|---|---|---|---|---|
| 1 | ETM+ | 131/38 | 2002-11-22 | 1% |
| 2 | ETM+ | 136/38 | 2003-1-28 | 0% |
| 3 | ETM+ | 132/41 | 2003-2-17 | 1% |
| 4 | TM | 136/33 | 2003-8-16 | 0% |
| 5 | TM | 141/35 | 2003-9-20 | 0% |
| 6 | TM | 135/33 | 2004-8-27 | 0% |
| 7 | TM | 137/39 | 2004-12-15 | 1% |
| 8 | TM | 132/38 | 2005-1-13 | 1% |
| 9 | TM | 136/39 | 2005-3-14 | 1% |
| 10 | TM | 132/34 | 2005-4-3 | 1% |
| 11 | TM | 142/34 | 2005-6-28 | 0% |
| 12 | TM | 136/36 | 2005-10-24 | 1% |
| 13 | TM | 133/38 | 2005-11-4 | 1% |
| 14 | TM | 135/39 | 2006-2-6 | 1% |
| 15 | TM | 135/33 | 2006-8-1 | 0% |
| 16 | TM | 141/39 | 2006-9-28 | 1% |
| 17 | TM | 132/34 | 2006-10-31 | 2% |
| 18 | TM | 134/37 | 2006-11-30 | 1% |
| 19 | TM | 136/39 | 2006-12-14 | 1% |
| 20 | TM | 134/33 | 2007-3-6 | 1% |
| 21 | TM | 141/34 | 2007-7-13 | 1% |
| 22 | TM | 139/35 | 2007-9-17 | 1% |
| 23 | TM | 132/37 | 2008-2-23 | 1% |
| 24 | TM | 132/42 | 2008-3-10 | 1% |
| 25 | TM | 134/36 | 2008-5-11 | 1% |
| 26 | TM | 145/35 | 2008-6-25 | 1% |
| 27 | TM | 142/34 | 2008-8-7 | 1% |
| 28 | TM | 150/33 | 2008-10-2 | 1% |
| 29 | TM | 130/37 | 2008-11-7 | 1% |
| 30 | TM | 133/37 | 2008-12-14 | 1% |
| 31 | TM | 132/38 | 2009-3-13 | 0% |
| 32 | TM | 132/37 | 2009-4-14 | 1% |
| 33 | TM | 147/35 | 2009-8-13 | 1% |
| 34 | TM | 151/33 | 2009-9-10 | 1% |
| 35 | TM | 138/37 | 2009-10-17 | 1% |
| 36 | TM | 134/36 | 2009-11-22 | 1% |
| 37 | TM | 133/38 | 2010-2-19 | 0% |
| 38 | TM | 135/39 | 2010-3-21 | 1% |
| 39 | TM | 150/32 | 2010-11-9 | 1% |
| 40 | TM | 134/39 | 2011-3-1 | 1% |
| 41 | TM | 141/35 | 2011-8-25 | 0% |
| 42 | TM | 132/37 | 2011-10-29 | 0% |
| 43 | OLI | 147/37 | 2013-4-18 | 2% |
| 44 | OLI | 149/34 | 2013-5-18 | 2% |
| 45 | OLI | 146/36 | 2013-8-1 | 1% |

| 46 | OLI | 145/36 | 2013-9-27 | 2% |
|----|-----|--------|-----------|-----|
| 47 | OLI | 141/35 | 2013-11-18 | 1% |
| 48 | OLI | 133/40 | 2014-1-13 | 1% |
| 49 | OLI | 136/38 | 2014-2-19 | 1% |
| 50 | OLI | 136/33 | 2014-7-13 | 0% |
| 51 | OLI | 144/35 | 2014-8-22 | 1% |
| 52 | OLI | 139/38 | 2015-1-10 | 1% |
| 53 | OLI | 143/39 | 2015-3-11 | 1% |
| 54 | OLI | 151/33 | 2015-10-13 | 1% |
| 55 | OLI | 136/38 | 2015-12-23 | 1% |
| 56 | OLI | 132/34 | 2016-5-3 | 2% |
| 57 | OLI | 151/33 | 2016-6-25 | 2% |
| 58 | OLI | 143/34 | 2016-9-21 | 1% |
| 59 | OLI | 133/37 | 2016-11-18 | 2% |
| 60 | OLI | 146/38 | 2017-2-1 | 1% |
| 61 | OLI | 151/33 | 2017-4-9 | 0% |
| 62 | OLI | 144/35 | 2017-7-29 | 1% |
| 63 | OLI | 133/37 | 2017-11-5 | 1% |
| 64 | OLI | 131/36 | 2018-4-16 | 1% |
| 65 | OLI | 133/38 | 2018-11-8 | 0% |
| 66 | OLI | 137/40 | 2018-12-22 | 0% |
| 67 | OLI | 131/38 | 2019-3-18 | 1% |
| 68 | OLI | 136/33 | 2019-8-28 | 0% |
| 69 | OLI | 151/33 | 2019-9-22 | 0% |
| 70 | OLI | 134/36 | 2019-11-2 | 1% |
| 71 | OLI | 132/38 | 2019-12-6 | 1% |
| 72 | OLI | 135/39 | 2020-1-12 | 1% |
| 73 | OLI | 136/38 | 2020-2-4 | 1% |
| 74 | OLI | 151/33 | 2020-8-23 | 1% |
| 75 | OLI | 149/35 | 2020-12-31 | 1% |
| 76 | OLI | 132/34 | 2021-1-25 | 1% |
| 77 | OLI | 143/34 | 2021-7-17 | 1% |
| 78 | OLI | 151/33 | 2021-9-27 | 0% |
| 79 | OLI | 135/36 | 2021-10-29 | 1% |
| 80 | OLI | 143/36 | 2021-11-22 | 0% |
| 81 | OLI | 151/33 | 2021-12-16 | 1% |

Why the validation accuracy of HMRF*solar* or HMRF*dem* is higher than MODIS? In my opinion, HMRF just filled the data gap, why the accuracy is also improved a lot. Please justify it.

**Response:** Our HMRF-based framework can exploit spatial and temporal contextual information and environmental association information, in addition to the MODIS spectral information that was used in the standard NASA algorithm to produce the original MODIS snow products. The category of all pixels (including data-gap pixels and gap-free pixels) on the entire initial MODIS snow cover products were determined by employing the optimal parameters and HMRF algorithm. As

demonstrated in our previous study (Huang et al., 2018), our HMRF framework not only fills the data gaps, but also improves the snow cover estimate accuracy of original MODIS snow cover products.

*Reference:*

Huang, Y., Liu, H., Yu, B., Wu, J., Kang, E. L., Xu, M., Wang, S., Klein, A., and Chen, Y.: Improving MODIS snow products with a HMRF-based spatio-temporal modeling technique in the Upper Rio Grande Basin, Remote Sensing of Environment, 204, 568-582, doi: 10.1016/j.rse.2017.10.001, 2018.

I have concerned that the weight used in Equation (1), such as $U_{xi}$, $U_{st}$, $U_{ev}$ are negative defined in Equation (3), (4) and (13).

**Response:** In Equation (1), $U_{xi}$, $U_{st}$, and $U_{ev}$ are the spectral, spatiotemporal, and environmental energy functions, respectively. Because the probabilities modeled by HMRF are equivalent to the energies characterized by a Gibbs random field (Geman and Geman, 1984), the maximization of the probability can be realized by minimizing total energy function (Huang et al., 2018, Equation (S1)):

$$\underset{C_2}{\text{Max}}\{P(\beta_n \,|x_i, N_{xi}, N_{st}, I_{ev})\} = \underset{C_2}{\text{Min}}\{\frac{1}{Z} e^{-[U_T(\,\beta_n, x_i, N_{xi}, N_{st}, I_{ev})]}\} \tag{S1}$$

where Z is a constant; $U_T$ is the total energy function, detailed derivation procedure and description can be found in Huang et al. (2018).

Thus, in previous Equation (3), (4), and (13) (now in Equation (4), (5), and (14)), $U_{xi}$, $U_{st}$, $U_{ev}$ are negative defined by using spectral probability, spatiotemporal probability, and environmental probability.

*Reference:*

Geman, S., Geman, D.: Stochastic relaxation, Gibbs distributions, and the Bayesian restoration of images. IEEE Transactions on Pattern Analysis and Machine Intelligence, PAMI-6(6), 721-741, 1984.

Huang, Y., Liu, H., Yu, B., Wu, J., Kang, E. L., Xu, M., Wang, S., Klein, A., and Chen, Y.: Improving MODIS snow products with a HMRF-based spatio-temporal modeling technique in the Upper Rio Grande Basin, Remote Sensing of Environment, 204, 568-582, doi: 10.1016/j.rse.2017.10.001, 2018.

Specific comments/suggestions:
Please provide the definition and equation of accuracy evaluation index (OA, OE et al.)

**Response:** The definition and equation of accuracy evaluation index have been provided in Line 270-278.

Figure 8. Please add the latitude and longitude information.
**Response:** The latitude and longitude have been added in Figure 8.

[Figure]

(a) Sentinel-2B        (b) Original MODIS SCP

(c) HMRF*ele*-based SCP        (d) HMRF*solar*-based SCP

Figure 8: Comparison between true-color Sentinel-2B imagery and(a), original MODIS snow products (b), HMRF$_{ele}$-based snow products (c) HMRF$_{solar}$-based snow products.

The Figure 4,5,7,8 resolution is too low, please check whether the Figure format meets the requirements of the journal.

**Response:** The resolution of Figure 4,5,7,8 have been improved

[Figure]

| Month | OA-HMRF*ele* | OA-HMRF*solar* | OE-HMRF*ele* | OE-HMRF*solar* | CE-HMRF*ele* | CE-HMRF*solar* | Snow cover |
|-------|-------------|----------------|--------------|----------------|--------------|----------------|------------|
| Sep | 87.62% | 90.02% (+2.40%) | 19.21% | 13.92% (-5.29%) | 4.37% | 3.09% (-1.28%) | 5.23% |
| Oct | 88.61% | 91.87% (+3.26%) | 20.76% | 14.53% (-6.23%) | 6.13% | 4.98% (-1.15%) | 7.75% |
| Nov | 87.26% | 90.03% (+1.91%) | 21.59% | 16.01% (-5.58%) | 4.09% | 3.19% (-0.90%) | 7.03% |
| Dec | 82.14% | 86.72% (+4.58%) | 21.84% | 18.37% (-3.47%) | 6.98% | 5.27% (-1.71%) | 10.30% |
| Jan | 90.29% | 92.17% (+1.88%) | 13.11% | 10.36% (-2.75%) | 6.15% | 3.59% (-2.56%) | 10.10% |
| Feb | 82.24% | 84.22% (+1.98%) | 25.49% | 21.29% (-4.20%) | 10.23% | 8.94% (-1.29%) | 13.32% |
| Mar | 89.03% | 90.78% (+1.75%) | 14.17% | 10.49% (-3.68%) | 12.82% | 10.17% (-2.65%) | 8.51% |
| Apr | 92.64% | 93.22% (+0.70%) | 17.64% | 14.72% (-2.92%) | 11.31% | 9.46% (-1.85%) | 6.49% |
| May | 92.32% | 94.40% (+2.08%) | 15.76% | 13.36% (-2.40%) | 10.31% | 8.29% (-2.02%) | 3.25% |
| Jun | 92.37% | 93.86% (+1.22%) | 12.38% | 10.39% (-1.99%) | 8.46% | 7.21% (-1.25%) | 1.30% |
| Jul | 90.53% | 91.07% (+2.69%) | 10.01% | 8.31% (-1.70%) | 2.33% | 2.14% (-0.19%) | 1.42% |
| Aug | 88.86% | 90.73% (+1.87%) | 14.15% | 12.82% (-1.33%) | 2.16% | 1.88% (-0.28%) | 2.13% |

Figure 4: Temporal variations in OA (a), OE (b), and CE (c) of HMRF$_{ele}$- and HMRF$_{solar}$-based snow products from 2002–2021.

[Figure]

| Elevation (m) | OA-HMRF*ele* | OA-HMRF*solar* | OE-HMRF*ele* | OE-HMRF*solar* | CE-HMRF*ele* | CE-HMRF*solar* |
|---|---|---|---|---|---|---|
| <3000m | 92.96% | 94.26% (+1.30%) | 27.31% | 20.98% (-6.33%) | 2.16% | 1.98% (-0.18%) |
| 3000–4000m | 89.15% | 91.33% (+2.18%) | 19.84% | 14.69% (-5.15%) | 7.23% | 5.91% (-1.32%) |
| 4001–5000m | 87.12% | 89.60% (+2.48%) | 13.24% | 9.77% (-3.47%) | 9.12% | 6.79% (-2.33%) |
| >5000m | 85.09% | 88.25% (+3.16%) | 20.37% | 16.12% (-4.25%) | 6.36% | 5.16% (-1.20%) |

Figure 5: Effect of elevation on OA (a), OE (b), and CE (c) of HMRF$_{ele}$- and HMRF$_{solar}$-based snow products from 2002–2021.

[Figure]

| Aspect | OA-HMRF*ele* | OA-HMRF*solar* | OE-HMRF*ele* | OE-HMRF*solar* | CE-HMRF*ele* | CE-HMRF*solar* |
|---|---|---|---|---|---|---|
| Sunny slope | 85.21% | 88.37% (+3.16%) | 22.96% | 19.33% (-3.63%) | 7.91% | 6.63% (-1.28%) |
| Shaded slope | 88.32% | 90.07% (+1.75%) | 18.47% | 17.10% (-1.37%) | 8.85% | 8.26% (-0.59%) |

Figure 7: Effect of aspect on OA (a), OE (b), and CE (c) of HMRF$_{ele}$- and HMRF$_{solar}$-based snow products from 2002–2021.

[Figure]

Figure 8: Comparison between true-color Sentinel-2B imagery and(a), original MODIS snow products (b), HMRF$_{ele}$-based snow products (c) HMRF$_{solar}$-based snow products.

Line115-116, "the values of 211, 237, and 239 in the NDSI_Snow_Cover_Class band were reclassified as non-snow". From #Line111, "the values of 211, 237 and 239" indicate "night time", "inland water", "ocean". So it is not reasonable that the pixels with three values are determined as non-snow.

**Response:** Thank you for your careful reading. In our data processing, the value of 211(night time) was determined as data-gap. As for the values of 237 (inland water) and 239 (ocean), we determined them as non-snow, as referenced as Huang et al. (2022). We have revised the classification scheme in Line 121.

*Reference:*

Huang, Y., Song, Z. C., Yang, H. X., Yu, B. L., Liu, H. X., Che, T., Chen, J., Wu, J. P., Shu, S., Peng, X. B., Zheng, Z. J., and Xu, J. H.: Snow cover detection in mid-latitude mountainous and polar regions using nighttime light data, Remote Sensing of Environment, 268, doi: 10.1016/j.rse.2021.112766, 2022.

---

## Author Response (AR2)

**Response to Reviewer 1**

To facilitate the readability of these revision notes for the editor and reviewers, the comments to the author are in blue, whereas the responses are in black.

**Comments to the Author:**

The revised paper reprocessed the input data for HMRF modeling and added more extended time series and terrain-corrected Landsat images for validation. The new accuracy assessment demonstrates the effectiveness and potential applications of the new snow cover dataset. Several minor comments still need to be solved before publication.

**Comment 1:**

In Figure 1, suggest changing the color of station symbols to more contrasting colors such as green, blue, etc. In addition, change 'Stations' to 'Meteorological station'

**Response:** We appreciate the reviewer's suggestion. In this revision, figure 1 has been modified as follows:

[Figure]

Figure 1: Topography, meteorological stations, and survey photos of the TP. (a) Surface elevation and distribution of meteorological stations in the TP. Landsat series data utilized for

validation and sample area are also shown. (b) and (c) in situ photos from field survey in the TP.

**Comment 2:**
Line 121: remove the comma after "237".
**Response:** Removed.

**Comment 3:**
Line 192-193: change the previous sentence "The general snow fraction estimation method was derived over other regions, and many studies have shown that the derived linear relationship has limited accuracy in the TP" to "The general linear relationship between FSC and NDSI was derived over other regions, which has limited accuracy in the TP", and remove "between FSC and NDSI" in Line193-194.
**Response:** Changed.

**Comment 4:**
Line 265: add "the" before "central pixel".
**Response:** Added.

**Comment 5:**
Line 281, 312: add a space before the number.
**Response:** Added.

**Comment 6:**
In Figure 8, the solar-based SCP compared to ele-based SCP, another area that has been significant improvement can be seen is in the upper left corner, where highlighting is recommended.
**Response:** As suggested, we have highlighted the significant improvement area in the upper left corner in Figure 8.

[Figure]

Figure 8: Comparison between true-color Sentinel-2B imagery and(a), original MODIS snow products (b), HMRF$_{ele}$-based snow products (c) HMRF$_{solar}$-based snow products.

**Comment 7:**
Line 415, 417, 466: change "omission error" and "commission error" into "OE" and "CE", respectively.
**Response:** We have changed the text as suggested in Line 392, 394, 441.

---

## Author Response (AR3)

**Response to Editor**

   To facilitate the readability of these revision notes for the editor and reviewers, the comments to the author are in blue, whereas the responses are in black.

**Comments to the Author:**

Thank you for your revision. I only have a minor comment here. I'm still concerned that it is correct that Inland water and ocean are considered as non-snow types, since there would be lake ice covered by snow for inland water frozen during winters.

**Response:** We appreciate the editor's suggestion. For version 6 MODIS snow products, the products user guide indicates that lake ice is included in the NDSI_Snow_Cover layer, and the lake ice or inland waters can be masked where their location is mapped in bit 0 of the NDSI_Snow_Cover_Algorithm_Flags_QA (Riggs et al., 2019). We also checked the classification scheme in other related snow studies (e.g., Chen et al., 2020; Huang et al., 2018; Huang et al., 2020; Li et al., 2020), the inland water and ocean are also reclassidied as non-snow types.

*References:*

Chen, S., Wang, X., Guo, H., Xie, P., Wang, J., and Hao, X.: A Conditional Probability Interpolation Method Based on a Space-Time Cube for MODIS Snow Cover Products Gap Filling, Remote Sensing, 12, https://doi.org/10.3390/rs12213577, 2020.

Huang, Y., Liu, H., Yu, B., Wu, J., Kang, E. L., Xu, M., Wang, S., Klein, A., and Chen, Y.: Improving MODIS snow products with a HMRF-based spatio-temporal modeling technique in the Upper Rio Grande Basin, Remote Sensing of Environment, 204, 568-582, https://doi.org/10.1016/j.rse.2017.10.001, 2018.

Huang, Y., Song, Z. C., Yang, H. X., Yu, B. L., Liu, H. X., Che, T., Chen, J., Wu, J. P., Shu, S., Peng, X. B., Zheng, Z. J., and Xu, J. H.: Snow cover detection in mid-latitude mountainous and polar regions using nighttime light data, Remote Sensing of Environment, 268, doi: 10.1016/j.rse.2021.112766, 2022.

Li, M., Zhu, X., Li, N., and Pan, Y.: Gap-Filling of a MODIS Normalized Difference Snow Index Product Based on the Similar Pixel Selecting Algorithm: A Case Study on the Qinghai–Tibetan Plateau, Remote Sensing, 12, https://doi.org/10.3390/rs12071077, 2020.

Riggs, G. A., Hall, D. K., and Román, M. O.: MODIS Snow Products Collection 6.1 User Guide, https://modis-snow-ice.gsfc.nasa.gov/uploads/snow_user_guide_C6.1_final_revised_april.pdf, 2019.